# Control of entropy in neural models of environmental state

**Timothy H Muller[1]\*, Rogier B Mars[1,2], Timothy E Behrens[1,3], Jill X O'Reilly[1,2,4]\***

[1]Wellcome Centre for Integrative Neuroimaging, Centre for Functional Magnetic Resonance Imaging of the Brain, University of Oxford, John Radcliffe Hospital, Oxford, United Kingdom; [2]Donders Institute for Brain, Cognition and Behaviour, Radboud University, Nijmegen, The Netherlands; [3]Wellcome Centre for Human Neuroimaging, Institute of Neurology, University College London, London, United Kingdom; [4]Department of Experimental Psychology, University of Oxford, Oxford, United Kingdom

**Abstract** Humans and animals construct internal models of their environment in order to select appropriate courses of action. The representation of uncertainty about the current state of the environment is a key feature of these models that controls the rate of learning as well as directly affecting choice behaviour. To maintain flexibility, given that uncertainty naturally decreases over time, most theoretical inference models include a dedicated mechanism to drive up model uncertainty. Here we probe the long-standing hypothesis that noradrenaline is involved in determining the uncertainty, or entropy, and thus flexibility, of neural models. Pupil diameter, which indexes neuromodulatory state including noradrenaline release, predicted increases (but not decreases) in entropy in a neural state model encoded in human medial orbitofrontal cortex, as measured using multivariate functional MRI. Activity in anterior cingulate cortex predicted pupil diameter. These results provide evidence for top-down, neuromodulatory control of entropy in neural state models.
DOI: https://doi.org/10.7554/eLife.39404.001

**\*For correspondence:**
timothymuller127@gmail.com (THM);
jill.oreilly@psy.ox.ac.uk (JXO'R)

## Introduction

Faced with a constant stream of complex sensory data, observers construct internal (mental) models of the underlying state of the world that generated that data. These models allow the observer to understand and make predictions about the environment. The current state of the environment can be thought of as a latent variable in such models, that is a variable that cannot be observed directly but must be inferred from sensory observations. For example a rat in a reversal learning task needs to know the underlying state of the world (which of two levers is more likely to deliver a food reward), but cannot observe the probabilities per se; instead the rat must infer the probabilities through repeated sampling of an observable variable (does the lever pay out on this trial or not?).

Theoretical accounts of inference (sometimes called learning, as in reinforcement learning) tend to include two distinct processes for updating beliefs over time. The first process is evidence-driven updating of beliefs – after each new piece of evidence (such as a reward) is observed, the model is updated to take into account the new observation. The second process is one that occurs *between* observations, as states of the world *for which no evidence has necessarily been observed* are up-weighted to account for the possibility that the environment *could* change before the next observation. Mathematically this second process is called the transition function as it determines the transition between posterior beliefs on trial t and prior beliefs on trial t + 1.

These two updating processes differ somewhat, both conceptually and algorithmically. Conceptually, evidence-driven updating, as the name suggests, is driven by recently

observed evidence and usually takes into account how that evidence differs from predictions under a prior belief (e.g., reward prediction error). In contrast, updating under the transition function is driven by higher order features of the environment such as the inferred level of environmental volatility and structural knowledge about which environmental states tend to follow each other (*Behrens et al., 2007*; *Meyniel and Dehaene, 2017*).

Algorithmic examples of evidence-driven updating would be the value update equation of the Rescorla-Wagner model, or the application of Bayes' theorem to combine a new observation with a prior belief. In contrast, a simple example of a transition function algorithm is a constant 'leak' that up-weights all possible future states of the environment equally, and down-weights beliefs based on past observations, such that the model's beliefs constantly decay towards a state of agnosticism (an example would be the leaky accumulator (*Usher and McClelland, 2001*)). A more complex transition function could determine structured changes in the environment, for example by preferentially predicting that if the environment changes, it changes to a similar state (an example would be the Gaussian transition function on reward probability (*Behrens et al., 2007*)).

Put another way, the overall level and distribution of uncertainty in an internal model is controlled in part by the transition function. When the environment is believed to be changeable (*Behrens et al., 2007*), or to have recently changed (*Wilson et al., 2010*), the transition function adjusts beliefs so that there is high uncertainty about the state of the world on trial t + 1, even given what was known at trial t. The optimal level of uncertainty is determined by two opposing forces: if relatively little uncertainty is added at each transition the model can utilize a longer history of observations to infer the current state of the world; but if too little uncertainty is added, the model becomes inflexible.

Previous studies have addressed the question of how the optimal level of uncertainty, and hence the magnitude of the 'leak' or transition function, should be determined by inferring the higher order statistics of the environment such as its volatility or change-point probabilities. Furthermore, blood-oxygen-level dependent (BOLD) and pupillometric signals tracking such (optimal) uncertainty have been observed (*Behrens et al., 2007*; *McGuire et al., 2014*; *Nassar et al., 2012*; *Preuschoff et al., 2011*). A second, open question, however, is how the brain gets from calculating how flexible one's beliefs should be, to controlling flexibility and uncertainty within a model that represents the current state of the world.

In relation to the first question, how the optimal level of uncertainty is calculated, Behrens and colleagues (*Behrens et al., 2007*) argued that the learning rate (analogous to the magnitude of the leak) should optimally depend on environmental volatility, and observed that when volatility (hence learning rate) was high, anterior cingulate cortex (ACC) was active. O'Reilly and colleagues (*O'Reilly et al., 2013*) showed activity in a similar region when participants were instructed that changes in the environment had occurred. McGuire and colleagues (*McGuire et al., 2014*) found activity in ACC associated with belief uncertainty, another model-based measure that should determine the magnitude of the transition function (in their case, the probability that the model abandons its prior beliefs entirely in favour of a flat prior).

In relation to the second question, how uncertainty/flexibility is implemented within internal models, there is less evidence, one reason being that it is difficult to measure the uncertainty of an internal neural model directly. An intriguing possibility is that increases in neural model uncertainty are controlled via a neuromodulatory route. In particular noradrenaline has been proposed as a candidate mechanism by which the level of uncertainty in neural models of the world may be controlled (*Aston-Jones and Cohen, 2005*; *Bouret and Sara, 2005*; *Iigaya, 2016*; *Martins and Froemke, 2015*; *Nassar et al., 2012*; *Preuschoff et al., 2011*; *Pulcu and Browning, 2017*; *Yu and Dayan, 2005*), and experimental manipulations increasing noradrenaline levels in cortex tend to lead to increased randomness in behavior (*Tervo et al., 2014*). One theoretical proposal, that noradrenaline performs a 'network reset' (*Bouret and Sara, 2005*), is particularly reminiscent of the second, transition-function updating process described above.

In the current experiment we focussed on the second question above. That is, we attempted to measure the level of uncertainty within a probabilistic world model, and determine signals that predicted changes in the level of this uncertainty. These are key components for any candidate mechanism for the implementation of a transition-function that increases uncertainty, such as a leak.

We first established that we could measure uncertainty in a neural representation of a simple task state space using functional MRI. Then we identified predictors of increases in belief uncertainty, in

the absence of overt changes in behaviour. In a simple experimental environment (4-arm bandit task), human participants formed a probabilistic model of the state of the environment (which bandit has the high pay-out rate) that could be decoded from the medial orbitofrontal cortex (mOFC). This model represented participants' certainty about the state of the environment in that the strength of the representation of the currently selected option, relative to other options, depended on the strength of evidence experienced, as captured by an optimal observer model. As well as measuring a neural model with fMRI, we simultaneously measured pupil dilation – an index of neuromodulatory state, commonly used as a measure of noradrenaline release to cortex. We found changes in pupil dilation predicted changes in uncertainty in the mOFC state model, providing evidence that the mechanism for driving up entropy in cortical models of the environment could be neuromodulatory. In circumstances when model entropy should increase, such as after observations likely to indicate a change in environmental state, activity in the ACC was observed, which in turn explained the strength of the pupil dilation response, suggesting ACC may be involved in regulating these neuro-modulatory processes.

## Results

To establish a task environment in which participants were constantly making inferences about the current state of the environment, we used a 4-arm bandit task (*Figure 1a*, *Figure 1—figure supplement 1*). At any given time, one of four response options had a high (70 or 90%) probability of delivering reward and the other three had a low (20%) probability. The high reward option switched after a variable number of trials (mean 20 ± SD 5). Participants' task was therefore to infer the current state of the environment, i.e. which option was the current high-reward option. Participants were explicitly instructed about the reward structure before beginning the task. Although there were only four options available for selection (denoted by a different colour), there were eight options on the screen at any given time (see Materials and methods for details and motivation). The individual trials in this task were presented in rapid succession to maximize design efficiency given that we were interested in a subset of trials.

Participants' behaviour alternated between exploitation and exploration as they formed and revised beliefs about the current state of the environment (*Figure 1c*). 72% of trials on average were classified as exploit trials (SEM across participants 1.2%); the mean length for an exploitation 'block' was 13.7 trials (across participants) and the within-subject standard deviation of exploitation block lengths was (mean across participants) 7.1 trials. In order to classify trials as either exploitative or exploratory, we used a simple heuristic. We defined the start of a period of exploitation as the first trial on which participants selected the true high reward probability option and received reward, and the end of the exploit period as the last trial before switching to a different option (thus within exploit periods, by definition, all choices were for the same option). The task therefore provided a clear behavioural marker of the onset of exploration, which was defined as the first trial on which participants switched away from making this selection. As an aside, in Appendix 1 we explore an alternative approach (*Ebitz et al., 2018*) to defining explore and exploit phases using a hidden Markov model (HMM). This approach yielded a very similar trial classification to the one we used. The relative merits of the two approaches are discussed in Appendix 1.

The possibility for voluntary exploration in the task created a constant pressure to evaluate whether the state of the environment had changed. Certainty about the state of the environment was modelled as the entropy across the state space in a Bayesian ideal observer model. The model, described in the Methods, had a space of possible states encompassing the possibility that each option was the current high reward option, and a range of possible pay-out rates; it also had a uniform leak.

We defined belief uncertainty as the entropy across the space of four possible states (namely, the four possible high-reward options, marginalizing over possible pay-out rates):

$$entropy = -\sum_i p(H_i) \cdot \log(p(H_i))$$

where $H_i$ is the state that option $i$ is the high reward option. For comparison we defined another measure of belief uncertainty, relative uncertainty (Wilson et al., 2010), which behaved similarly and

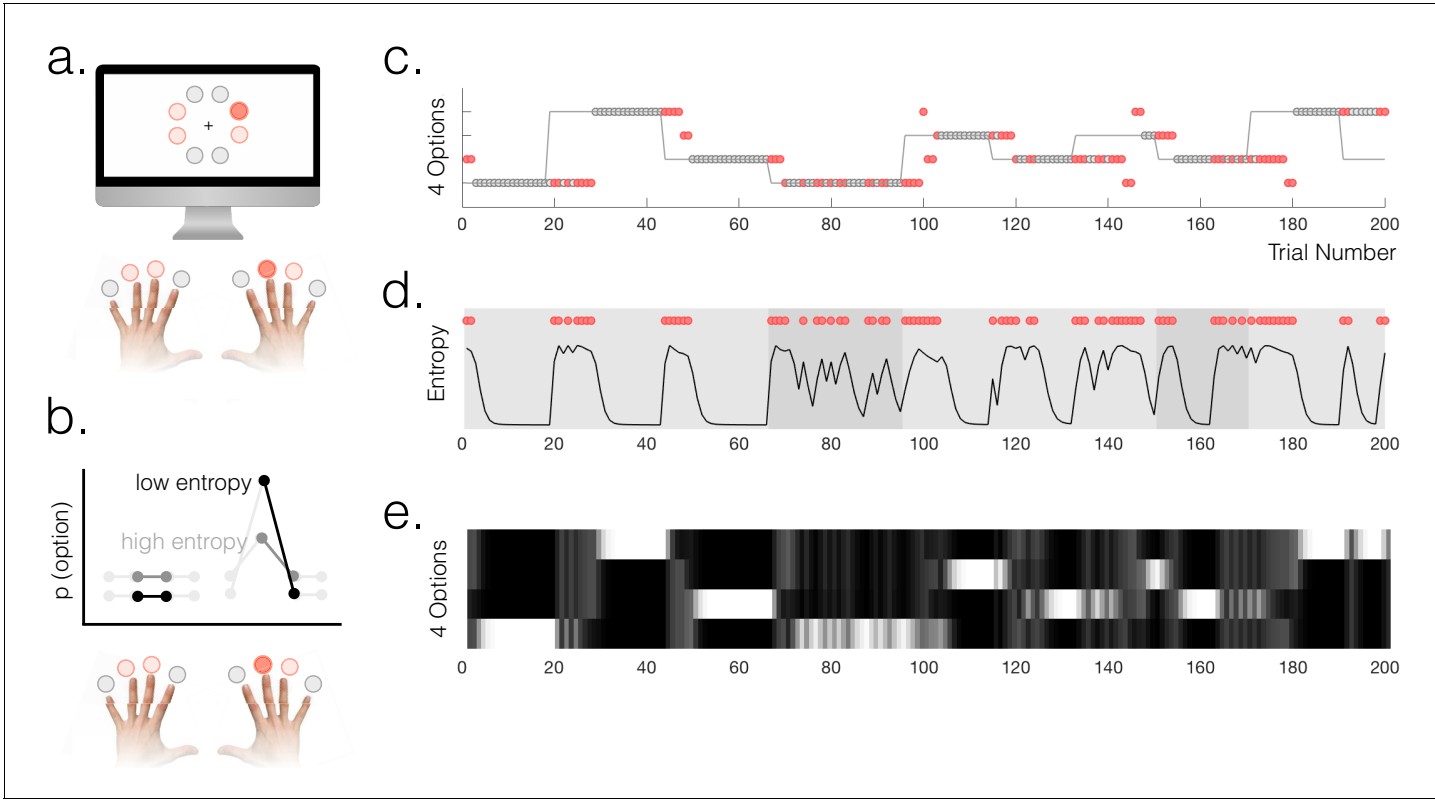

**Figure 1.** Task and behaviour. (**a**) Participants chose freely between four available options on each trial. A total of 8 options were used in the experiment but only four were used in each run of the task (indicated by option colours: red circles above the fingers denote the available options, and the dark red circle denotes the selected option). (**b**) Schematic of a probabilistic model of the state of the environment – the weighting of each option represents its probability of being the high reward option. When entropy is low, i.e. participants are certain, the weighting of the selected option is higher than when entropy is high. (**c**) Example behaviour of a participant. Dot markers denote choices (location in y-dimension indicates which option was chosen) – grey circles are rewarded and red dots are unrewarded trials. The grey line indicates the true state of the world (high reward option), which the participant must infer. Note alternating phases of exploration and exploitation – the main analyses presented in the paper refer only to the exploitation phase in which there are no overt changes of action. (**d**) Model entropy for the same run as in (**c**). Note that model entropy is low during exploit periods but increases following a reward omission (red dots are trials with reward omission). Background shading indicates the pay-out rate of the high reward option: during low pay-out (70%) exploit periods, indicated in dark grey, entropy tends to remain higher; during high pay-out periods (light grey) entropy reaches a floor during exploitation. (**e**) State space of the model – four horizontal tracks represent the four possible states of the environment; shading indicates the posterior probability assigned to each state by the model (light colours indicate high probability).

DOI: https://doi.org/10.7554/eLife.39404.002

The following figure supplements are available for figure 1:

**Figure supplement 1.** Implementation of the task described in the Main Text and Methods.

DOI: https://doi.org/10.7554/eLife.39404.003

**Figure supplement 2.** Exploitation and exploration engage different brain regions.

DOI: https://doi.org/10.7554/eLife.39404.004

**Figure supplement 3.** Activation related to task factors.

DOI: https://doi.org/10.7554/eLife.39404.005

**Figure supplement 4.** A large amount of the difference between exploitation and exploration is captured by task factors.

DOI: https://doi.org/10.7554/eLife.39404.006

was highly correlated with entropy (mean correlation across participants r=0.80). The relation between entropy and relative uncertainty is explored further in Appendix 1.

Model entropy was generally high in the exploit phases and low in the explore phases of the task (*Figure 1d,e*); it also tended to be higher when the pay-out rate of the high reward option was lower. Both high entropy, and high relative uncertainty, made it more likely that participants would switch from exploitation to exploration following a reward omission (two separate logistic regressions: entropy - $t_{18}$ = 7.60, p=2.57×10$^{-7}$; relative uncertainty - $t_{18}$ = 13.50, p=3.7×10$^{-11}$).

Furthermore, using an action policy defined by fitting two multinomial softmax functions (one for explore trials, one for exploit) to the option probabilities $p(H_i)$, we found that the particpants' choices were correctly predicted on 95% of exploit trials and 51% of explore trials (chance would be 25%). The importance of using a model for the study was in fact to have a representation of participants' latent belief states during phases when their overt behaviour remained stable, rather than to predict overt changes of behaviour; however the fact that the model could predict behaviour relatively well across all phases of the task speaks to its validity.

The overall distribution of brain activity in explore and exploit phases of the task differed (GLM1, see *Figure 1—figure supplement 2*). The explore phase engaged a frontal-parietal action network and the dorsomedial prefrontal cortex (dorsal anterior cingulate dACC and adjacent preSMA). The exploit phase engaged medial orbitofrontal cortex (mOFC) and the hippocampus, two regions known to be engaged in model-based choice (*Boorman et al., 2016*; *Jones et al., 2012*; *Noonan et al., 2010*; *Wikenheiser et al., 2017*). In addition to state uncertainty (model entropy), several factors differed between explore and exploit phases, notably the frequency of response switching, and the frequency of reward omission. Further analysis (GLM2, *Figure 1—figure supplement 3*) indicated that response switching and high model entropy (see below for details of model) predicted activity throughout the frontal parietal network and dorsomedial frontal cortex. Reward omission predicted activity in dorsomedial frontal cortex only, and reward activated ventral striatum. Low model entropy predicted activation of medial OFC and hippocampus (*Figure 1—figure supplement 3*). Most of the differences in activity observed between exploitation and exploration were explained by these factors (GLM3; *Figure 1—figure supplement 4*).

However, the main focus of our analysis was the variation in state uncertainty within the exploitation phase of the task, during which the chosen action was by definition constant, and how this uncertainty may be represented and modulated. By limiting our analysis to core exploitation trials (at least five trials, and therefore approximately twenty seconds, after the last action switch and at least five trials before the next action switch; such trials were 25% of all trials (mean across participants = 201/800 trials, SEM = 7.0)), we were able to identify neural activity concerned with changes in participants' beliefs in the absence of changes in overt behaviour. This dissociated uncertainty about the state of the environment from variability of action. All analyses presented are limited to this subset of trials unless otherwise stated.

During each exploitation block, participants' certainty about the state of the environment varied due to probabilistic reward omission. To generate trial-by-trial estimates of the certainty of participants' beliefs to regress against neural responses, we constructed a normative Bayesian learning model (see Materials and methods). The model inferred the trial-wise probability that each of the four options was the high reward probability option, based on the data observed by each individual participant (*Figure 1e*). For each trial, we calculated the entropy of the model's posterior probability distribution over options, based on evidence up to and including that trial (*Figure 1d*). This was used as a proxy for the entropy in the participants' internal model, at the point of feedback on each trial. Entropy is low when the probability mass of the posterior is mostly over one option, i.e. participants are certain. Within exploitation phases, model entropy generally decreased over time as participants continued exploiting a single option, but increased following probabilistic omissions of reward, and accordingly depended on the payout rate of the high reward option (70% or 90% reward) (see *Figure 1d,e*).

Throughout the next sections of this paper, we discuss the relationship between model entropy (in our theoretical model) and the level of uncertainty in a neural model of the state of the world. However, it should be noted that in the current task, the possible states of the environment are simply the alternatives for which option is the high reward option, that is {option A is high reward option... option D is high reward option}. Thus, in the current task, state and option value are inherently interlinked. It is necessarily the case that high confidence in the currently chosen option (low entropy state) correlates with that option having a high expected value (mean correlation ± s.e.m: -0.93 ± 0.005). Thus it should be noted that throughout the subsequent analyses, a low entropy trial, defined by us as a trial on which one particular option is considered very likely to be the high reward option, could equivalently be described as a state in which one particular option (the selected one) has a high expected value and others have a low expected value. In behavioural control more generally, this association would hold for any world model for which the aim was to determine the expected value associated with different candidate actions.

## Decoding a probabilistic representation of beliefs about the state of the environment

A probabilistic model of the current state of the world should comprise representations of candidate states, weighted by their probabilities. Recent studies have identified medial OFC as a likely site for such a probabilistic model in the context of model-based reinforcement learning (*Chan et al., 2016*; *Schuck et al., 2016*; *Wilson et al., 2014*).

In the present task, such a probabilistic internal model could be thought of as a mixture model comprising representations of the possible options, wherein the strength of the representation of a given option on each trial is weighted by the belief that option is the high reward probability option (*Figure 1b*).

To identify regions that represented the task environment in this probabilistic manner, we used a multivariate searchlight approach over the whole brain as follows: at each searchlight, in one half of the data (training set) we identified activity patterns characteristic of trials on which each option was selected. In the other half of the data (test set) we calculated the trial-wise probability of classifying as each option (we then swapped training and test sets; see Materials and methods for details).

Simply searching for regions in which we could correctly classify the selected option, we were, as expected, able to decode at above-chance levels from a broad network of areas including motor and visual cortex (*Figure 2a*). A plot of the probability of decoding each action from visual cortex (*Figure 2c*, top) and motor cortex (*Figure 2c*, middle) reveals a representational structure that is sensitive to spatial and motor factors; the most likely options to be decoded, other than the actually selected option, are those corresponding to the chosen hand (and equivalently, the chosen visual hemifield).

However, the defining feature of a probabilistic model is that the relative weighting of the chosen option should depend on certainty, as shown in schematic *Figure 1b*. To search for regions that had this property, we defined a trial-wise summary measure of representation strength as the ratio of the odds of classifying the trial as the chosen option, to the odds of classifying it as one of the unchosen options. This odds-ratio measure captured the extent to which the chosen option was represented more strongly than unchosen alternatives on each trial. Finally, we regressed trial-by-trial model entropy (as a proxy for the entropy of participants' beliefs) against representation strength (GLM4). This identified areas in which representation strength for the chosen option was higher when there was higher certainty that the chosen option was the high reward option.

Using this approach we identified a region in mOFC (peak = 10,40,–22; *Figure 2b*) in which representation strength for the exploited option was higher when model entropy was lower (corrected for multiple comparisons using permutation testing; cluster mass p<0.05 corrected with a cluster forming threshold of p=0.001). This was the only cluster to survive multiple comparisons correction. The same effect was also detectable in the hippocampus ($t_{(18)}$=-2.11 p<0.05 uncorrected at an ROI centered on the peak voxel 34, -14, -16 for the exploit vs. explore univariate contrast from GLM 1; *Figure 1—figure supplement 2*). However the effect in hippocampus did not survive whole brain correction for multiple comparisons and should therefore be treated with caution.

The weighted representation of option preferences observed in mOFC is consistent with a probabilistic internal model of the environment, whereby the currently selected option has a higher decoding probability when model entropy is low (*Figure 1b* and *Figure 2c*, bottom). It is distinct from an action/policy representation, in which the representation of options is not modulated by entropy, as was observed in visual and motor areas (*Figure 2c*, top and middle).

To probe the robustness of the mOFC result, we repeated the analysis using two measures other than the model entropy regressor to predict representation strength. Entropy is a summary measure of the uncertainty across options; low entropy could arise from any of the four options being represented more strongly than the others. To check that the result was specific to the current state of the environment, we repeated the above analysis but instead of model entropy, used the probability (belief) assigned by the model that the currently exploited option is the high reward option as the predictor variable (GLM5). This also produced a specific effect in the medial OFC (*Figure 2—figure supplement 1*). To confirm our result was not dependent on the particular learning model we used, we simply compared representation strength during exploit periods with high and low payout rates. When the pay-out rate of the exploited, high reward option was 90% vs. 70%, representation strength was higher in mOFC (*Figure 2—figure supplement 1*).

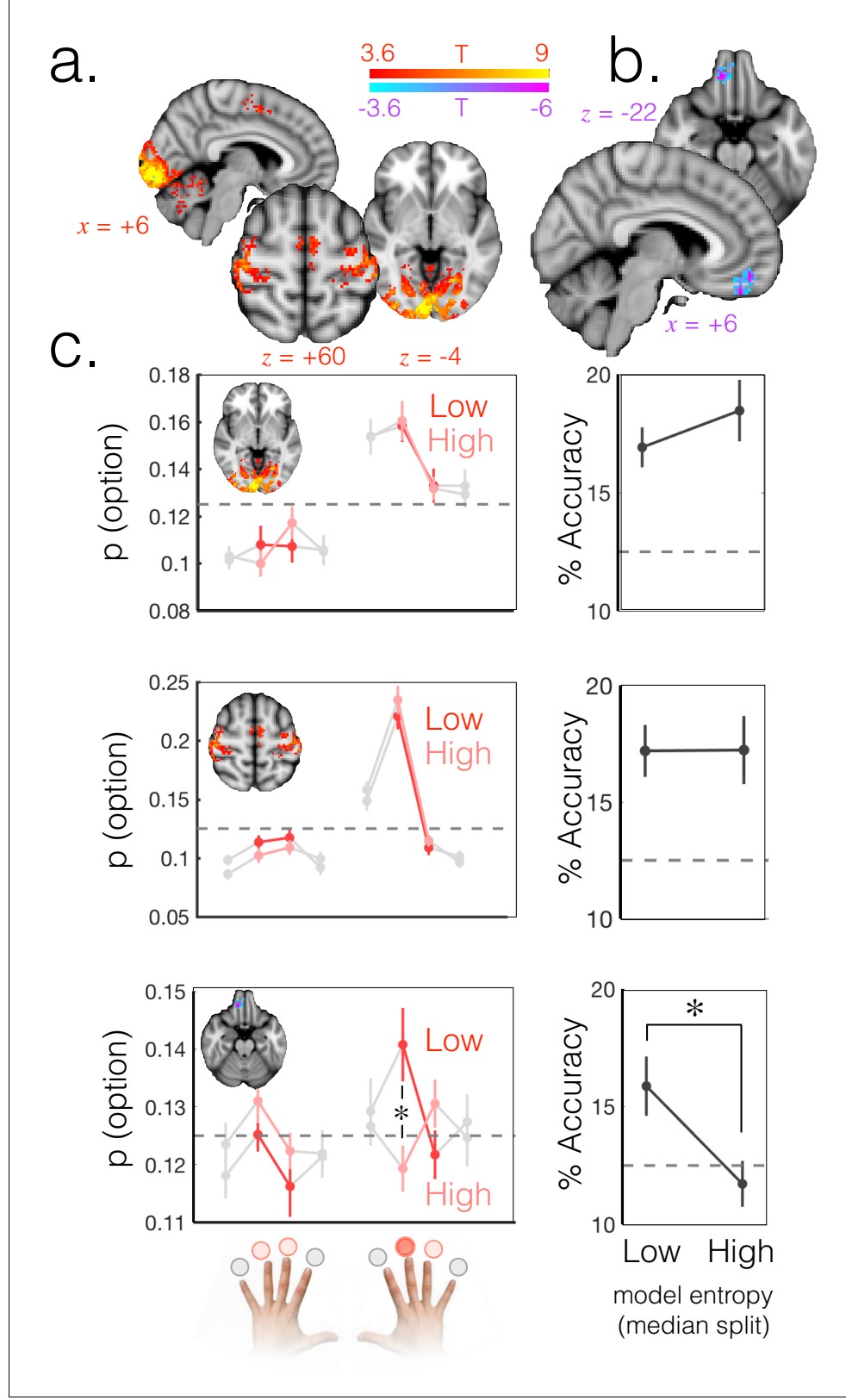

**Figure 2.** Probabilistic beliefs represented in mOFC. (**a**) The currently selected option can be decoded above chance, as expected, in motor and visual cortex (t score map for above chance decoding of the chosen option across subjects; thresholded at p<0.001; corrected for multiple comparisons). (**b**) Medial OFC represents the

*Figure 2 continued on next page*

*Figure 2 continued*

current state of the task probabilistically. Highlighted voxels are the centers of searchlights in which the representation strength for the chosen option was higher when model entropy was low (i.e. when participants have high certainty about the state of the environment; t score map for the effect of model entropy on representation strength; thresholded at p<0.001; corrected for multiple comparisons). (c) Region of interest analyses demonstrating multivariate decoding patterns. Left column: the multivariate classifier probability that each option was selected; on trials when model entropy was low (dark red) and high (light red). Right column: decoding accuracy (i.e. proportion of trials when the option with the highest probability of having been selected by the classifier was indeed the option selected by the participant) in low and high entropy trials. All error bars are SEM. Dashed lines denote chance. Top row and middle row: decoding in visual and motor cortices, respectively, is not sensitive to model entropy and errors in decoding tend to be to neighbouring options. Bottom row: decoding in mOFC is modulated by model entropy, such that decoding is higher when model entropy is low.

DOI: https://doi.org/10.7554/eLife.39404.007

The following figure supplements are available for figure 2:

**Figure supplement 1.** Representation strength in mOFC is explained by probability assigned to the currently selected option, as well as the difference between high and low reward exploit periods.

DOI: https://doi.org/10.7554/eLife.39404.008

**Figure supplement 2.** Histogram of t scores for the effect of entropy on representation strength (GLM4) for null data produced by shuffling voxel identities prior to PCA.

DOI: https://doi.org/10.7554/eLife.39404.009

## What is the nature of the representation in mOFC?

The multivariate whole brain analysis showed that the identity of the chosen option could be decoded with greater confidence (representation strength was higher) when model entropy was low. Whilst OFC has been proposed as the site of a representation of the current state of the world (*Schuck et al., 2016*; *Wilson et al., 2014*), it is also concerned with value representation. Trials with low model entropy in our task tended to be the same trials on which a high value option was chosen (an option which the participant believed had a high probability of leading to a reward), and trials on which recent reward history contained few reward omissions.

The reason we interpret the multivariate results as evidence for a probabilistic representation of the current state (which option is the current high reward option) is that our ability to decode the *identity* of the chosen option from mOFC (indexed as representation strength for the *identity* of that option) was higher when the participant's uncertainty (model entropy) was lower. Therefore decoding is driven by the relative weightings of representations of specific options rather than decoding a general value representation. Indeed, the training and test sets used in the analysis both included cases where the option value was higher and lower, which was unknown to the classifier. We examined whether our effects still held after regressing out a univariate effect of option value by repeating the analysis after first regressing out the effect of value from univariate activity in each voxel in the mOFC region of interest. The effect of model entropy on representation strength remained significant ($t_{(18)}$=-7.62,p<0.001). This was also true when regressing out recent reward history (whether the current and previous three trials were rewarded) in addition to value ($t_{(18)}$=-7.26,p<0.001) from voxelwise activity before carrying out multivariate analysis. Since these analyses were performed on the residual voxelwise data after the effect of value (or reward history) on voxelwise activity was removed, univariate option value could not drive the classification. Regardless, given the decoding effects were on specific option *identities*, decoding a general value representation cannot explain our results. Rather the decoding must be driven by different relative weightings of specific option representations, consistent with a probabilistic state representation.

We should reiterate here that in the current task, state and option value are inherently interlinked because the possible states of the environment are simply the alternatives for which is the high reward option, i.e. {option A is high reward option... option D is high reward option}. Whilst it is the representation strength of state *identity* that depends on uncertainty, the states themselves are defined in terms of option value. Therefore the present results in no way speak against the hypothesis that mOFC represents option values; in fact the opposite is true.

# A candidate neuromodulatory mechanism for increasing flexibility of belief representations

Having confirmed that a probabilistic model of the state of the world is represented in the brain, we went on to ask how entropy in that model might be controlled. An intriguing possibility is that increases in neural model uncertainty are controlled via a neuromodulatory route. The neuromodulator noradrenaline has been proposed as a candidate mechanism by which the level of uncertainty in neural models of the world may be controlled (*Aston-Jones and Cohen, 2005*; *Bouret and Sara, 2005*; *Iigaya, 2016*; *Martins and Froemke, 2015*; *Nassar et al., 2012*; *Preuschoff et al., 2011*; *Pulcu and Browning, 2017*; *Yu and Dayan, 2005*). Noradrenaline is produced in the locus coeruleus (LC), and is released very broadly in the cerebral cortex including the mOFC (*Aston-Jones and Cohen, 2005*).

We used baseline pupil size (mean pupil area in the 20 ms before outcome presentation, expressed as % signal change relative to the participant's mean pupil area over the task run) to index neuromodulatory state. Pupil size has primarily been interpreted in the literature as indexing noradrenaline levels, although the muscles controlling pupil size are sensitive to both noradrenaline and acetylcholine. Changes in pupil diameter have been linked to single-unit activity in the (noradrenergic) LC (*Joshi et al., 2016*). Recent physiological work has demonstrated that both noradrenaline and acetylcholine as measured in cortex, are associated with pupil size (*Reimer et al., 2016*); levels of both noradrenaline and acetylcholine were correlated with pupil dilation over a time course of up to a few seconds (a similar time course to the interval between trials in the current experiment). Therefore our pupillometry results cannot be interpreted as specific to noradrenaline but rather reflect a complex combination of neuromodulatory and other factors. However, *Reimer et al. (2016)* also report that noradrenaline, but not acetylcholine, was correlated with changes in pupil dilation (the temporal derivative of pupil size) as well as pupil size itself, which favours a noradrenergic interpretation of our pupillometry results (most of the pupillometry results reported below concern changes in pupil size rather than pupil size per se). However, we must emphasise that pupillometry is a very indirect measure of neuromodulation and although we mostly interpret our results in the context of noradrenaline due to the extensive literature on pupillometry and noradrenaline, as well as influential theories and experimental work linking noradrenaline to uncertainty, model updating and exploration, we note that our pupil size data cannot be specifically linked to noradrenaline over other pupil-linked arousal factors such as acetylcholine.

Baseline pupil size was larger in explore than exploit trials (*Figure 3a*, left), and increased when transitioning into exploration (*Figure 3—figure supplement 1*). On a trial-by-trial basis, reward omission produced a sustained increase in pupil size following feedback (*Figure 3a*, right).

We hypothesized that pupil dilation would predict changes in the level of uncertainty in the internal model. To test this hypothesis, we carried out a linear regression (GLM6) in which the change in baseline pupil size from one trial to the next was used to predict changes in representation strength in the mOFC from one trial to the next. In other words, we asked whether change in pupil size induced by observation of an outcome predicted decreases in representation strength between that trial and the next, providing evidence for changes in neuromodulatory state applying something analogous to a leak to the posterior belief or partial network reset (*Bouret and Sara, 2005*). The OFC representation strength was extracted from a region of interest (searchlight center) defined on the peak result shown in *Figure 2b* (change in model entropy was included as a co-regressor of no interest).

This analysis indicated that changes in baseline pupil size indeed explained changes in representation strength in mOFC; as expected, this relationship was negative, such that increases in pupil size predicted decreases in representation strength. Interestingly, this relationship was limited to unrewarded trials ($t_{(15)}$=-2.94, p=0.01); within the subset of rewarded trials there was no relationship ($t_{(15)}$=0.37, p=0.71; difference between rewarded and unrewarded: $t_{(15)}$=2.82, p=0.01; *Figure 3b*). This is significant because (due to the 4-arm nature of the task and the fact that we included only exploitation trials in the analysis) increases in uncertainty occurred almost exclusively following reward omissions – on only 0.2% of core exploit trials (10/3833 trials over all participants) did uncertainty increase following a rewarded trial. Therefore the relationship between pupil dilation and change in representation strength seems to hold in cases where uncertainty is increasing, but not when it is decreasing. To test the temporal specificity of the result, we shifted the pupil and OFC

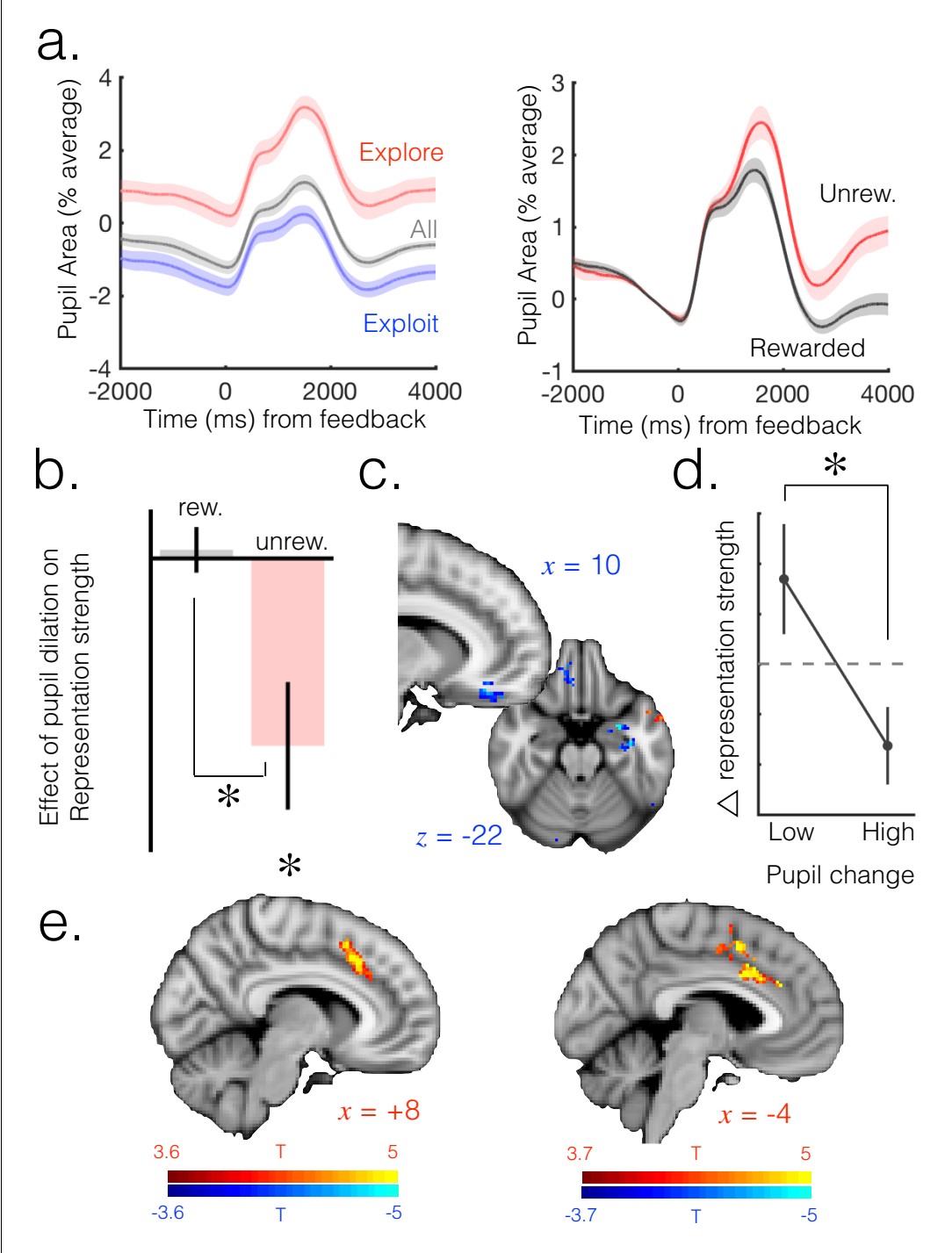

**Figure 3.** Neuromodulatory systems as a candidate mechanism for increasing flexibility of belief representations. (a) Pupil size mean timecourses throughout a trial. Left: mean timecourses shown for all trials as well as trials split according to whether they were explore or exploit trials, revealing a larger pupil size in exploration than exploitation. Right: splitting trials according to whether the trial was rewarded or not reveals a sustained increase in pupil size following omission of reward. Shaded regions denote SEM. (b) Change in pupil diameter predicts changes in representation strength in mOFC on exploit trials. Beta weights for the effect of change in pupil diameter on change in representation strength are significantly below zero. This is true only on trials when reward was omitted. Error bars are SEM. (c) Performing this analysis as a whole brain reveals, again, a relatively localized effect in mOFC (t score map shown, thresholded at p<0.01, uncorrected; analysis performed on unrewarded exploit trials). (d) Median split on unrewarded exploit trials reveals that on trials when pupil change is high, change in representation strength is more negative than when pupil change is lower, in mOFC. Error bars are SEM. (e) Left: ACC region active when model entropy increased. Right: ACC region in which activity predicted changes in pupil

*Figure 3 continued on next page*

*Figure 3 continued*

dilation, over and above the effect of increase in model entropy and mean brain activity. Both thresholded at p<0.001 and corrected for multiple comparisons.

DOI: https://doi.org/10.7554/eLife.39404.010

The following figure supplements are available for figure 3:

**Figure supplement 1.** Task-related pupil size changes are not explained by outcome stimulus type.

DOI: https://doi.org/10.7554/eLife.39404.012

**Figure supplement 2.** The relationship between change in baseline pupil diameter and change in representation strength in mOFC replicates in explore trials.

DOI: https://doi.org/10.7554/eLife.39404.014

**Figure supplement 3.** ACC activity explains changes in baseline pupil size across all trials.

DOI: https://doi.org/10.7554/eLife.39404.015

**Figure supplement 4.** ACC is engaged when transitioning from exploitation to exploration.

DOI: https://doi.org/10.7554/eLife.39404.016

**Figure supplement 5.** Individual pupil effects.

DOI: https://doi.org/10.7554/eLife.39404.011

**Figure supplement 6.** Breaking central fixation does not alter pupil effects.

DOI: https://doi.org/10.7554/eLife.39404.013

signals relative to one another and recomputed the effect. We found that the result was no longer significant when shifting the signals in any direction (p>0.1 for all analyses), suggesting the effect was temporally specific to change in pupil size predicting change in representation strength from the current trial to the next. This effect remained significant when regressing out value from the univariate signal before fitting the multivariate model, as discussed above (unrewarded: $t_{(15)}$=-3.07, p<0.01; rewarded: $t_{(15)}$=0.56, p=0.58; difference: $t_{(15)}$=2.95, p<0.01).

Although we focus here on the exploitation phase, we found this result replicated in the mOFC ROI when including only explore trials in the analysis (*Figure 3—figure supplement 2*; unrewarded: $t_{(15)}$=-2.38, p=0.03; rewarded: $t_{(15)}$=0.97, p=0.35; difference: $t_{(15)}$=-2.23, p=0.04). Furthermore, performing the analysis on unrewarded explore trials over the whole brain reveals a region relatively localized to mOFC (*Figure 3c*). To illustrate this result further, we took the unrewarded exploit trials and looked at the change in representation strength in the mOFC ROI on trials separated according to a median split on change in pupil diameter. As expected, we found change in representation strength was more negative when change in pupil dilation was higher (*Figure 3d*; $t_{(15)}$ = 2.72, p=0.02).

These results demonstrate that on trials when evidence against the current belief is observed (unrewarded trials), the change in uncertainty in the neural model is predicted by changes in pupil diameter, suggesting a role for neuromodulation in control of uncertainty in neural models. The fact the relationship is specific to unrewarded trials suggests that the relationship between pupil dilation and representation strength in mOFC is directionally specific to the process of increasing, not decreasing, model entropy.

## Anterior cingulate cortex is activated when model entropy increases and predicts pupil change

Having provided evidence that neural model uncertainty may be increased via pupil-linked neuromodulatory mechanisms, we next asked which brain regions are engaged during increases in uncertainty and may regulate changes in neuromodulatory systems. We used a whole brain GLM analysis to identify regions that were sensitive to increases in model entropy, even in the absence of overt action switches (analysis again limited to exploit trials as defined above; GLM7). This analysis identified activity in the ACC and adjacent preSMA that was directionally sensitive to change in entropy (*Figure 3e*, left). The greater the increase in entropy from the previous to current trial, the greater the activity in the ACC and surrounding regions of the dmPFC – that is, these regions were more active when entropy was increasing and less active when it was decreasing. These results are consistent with the observation ACC has been associated with the processing of feedback that should lead to changes of belief (*O'Reilly et al., 2013*).

ACC is one of the few cortical regions reported to have an afferent influence on the noradrenergic LC (*Jodo et al., 1998*) . This provides at least one anatomical route by which, if ACC is responsible for computing changes in beliefs, it may be able to broadcast increases in uncertainty to other brain regions via a neuromodulatory route. To probe the relationship between ACC and pupillometric effects, we asked where in the brain univariate activity predicts changes in baseline pupil size. We conducted a whole brain GLM analysis, again focused on the exploit period, in which univariate brain activity at each voxel was the predictor variable and change in pupil size the dependent variable; change in model entropy was included as a co-regressor. Pupil dilation is globally correlated with neural gain (*Eldar et al., 2013*), and we observed a strong correlation of pupil size with mean BOLD signal across the entire grey matter. When we controlled for this global effect by including mean activity across the grey matter as a co-regressor of no interest (GLM8), we observed a localized effect in ACC such that ACC activity on a given trial predicted the change in baseline pupil size from that trial to the next (*Figure 3e*, right; more activity in the brain associated with a greater increase in pupil dilation; corrected for multiple comparisons using permutation testing; p<0.05 at cluster forming threshold of p=0.001). Although we focus on the exploit period, this result is also present when conducting the analysis across all trials (*Figure 3—figure supplement 3*).

Given that ACC activity predicts pupil dilation and pupil dilation predicts changes in representation strength in mOFC, we tested whether there was a direct relationship between ACC activity and changes in representation strength in mOFC. However, there was no evidence for such a relationship (activity extracted from an ROI defined on peak of the ACC effect from either GLM7 or GLM8 did not have a significant relationship with change in representation strength in mOFC; p>0.1 for all trial subsets tested). This may reflect the fact that both brain measures have relatively low SNR compared to the pupillometry data, to which both were significantly related.

Although we focus on the exploitation phase in which overt behavior was stable, it is worth noting that a very similar pattern of activity in medial prefrontal cortex was present when participants switched from exploitation to exploration (GLM2, *Figure 3—figure supplement 4*). The activity observed in ACC in this analysis is activity over and above that driven by changing response/option selection (GLM2), which (by definition) occurs on the last trial of exploitation. Note that this effect is specific to exploration initiation, and was not observed when switching strategy from exploitation to exploration (*Figure 3—figure supplement 4*), suggesting a particular role in disengaging from the exploited strategy.

## Discussion

We set out to investigate the mechanisms by which entropy in neural models of the world may be controlled, particularly mechanisms by which entropy may be increased. We identified patterns of activity indicative of a probabilistic model of the environment in the medial OFC. The hallmark of this probabilistic model is that it has stronger representation strength for the currently selected option when model entropy is low. This is distinct from sensory and motoric representations in visual and motor cortex that are not sensitive to model entropy. Changes in representation strength in medial OFC, over and above those predicted by our task model, were explained by change in pupil dilation, a proxy measure for neuromodulator activity. Pupil dilation was in turn predicted by activity in the ACC – a region also active when model entropy increased.

That an internal model of the task is represented in mOFC is consistent with the proposal that this region represents the current location in task state space (*Wilson et al., 2014*), and in particular represents information that is hidden to participants but required to solve the task (*Chan et al., 2016*; *Schuck et al., 2016*). Our work extends this proposal to include the representation of the strength of dynamic beliefs in a probabilistic task in which participants infer the hidden option-reward contingencies that define their location in the task state space.

The fact that mOFC is the locus of the probabilistic model in the current task is probably due to the choice of task, as mOFC plays a key role in representing stimulus-action-outcome contingencies during model-based action (*Jones et al., 2012*). Different stimulus or task domains could rely on probabilistic models represented in other, task relevant parts of cortex. Furthermore, the exact nature of the representation we are interrogating in mOFC is an interesting avenue for future research. For example, it is possible the representation we observe here reflects the online representation of option-reward contingencies required to guide choice – for example, for online comparison

– and is not where the option-reward contingencies are stored offline, which may be in another region such as hippocampus (*Boorman et al., 2016*) or striatum (*Packard and Knowlton, 2002*). The hippocampus showed some limited evidence of probabilistic option representations in the current study. The striatum also plays a key role in representing action-outcome associations, and indeed mechanisms for maintaining flexibility within these representations have been proposed (*Franklin and Frank, 2015*). In the current study we were unable to detect evidence of a probabilistic model in the striatum. This could be because our multivariate analysis relied upon patterns across many voxels and was therefore relatively less sensitive to effects in small structures. Regardless of its exact nature, we are able to use the probabilistic model of the task space detected in mOFC as an index of the neural representation of participants' beliefs; that is, a neural measure of an internal model.

We next sought evidence for a role of neuromodulatory systems in increasing uncertainty in such an internal model. Although most theoretical work in this field has focussed on noradrenaline, it should be noted that our results concern pupil dilation, not noradrenaline per se. Pupil dilation depends on at least two neuromodulators (noradrenaline and acetylcholine) (*Joshi et al., 2016*; *Reimer et al., 2016*), as well as other arousal-related factors, so our results cannot be linked directly to noradrenaline.

A large body of theoretical work has ascribed noradrenaline various functions relating to the modification of internal models (changing gain (*Aston-Jones and Cohen, 2005*), network resets (*Bouret and Sara, 2005*), and signalling unexpected uncertainty (*Preuschoff et al., 2011*; *Yu and Dayan, 2005*)). However, evidence for the impact of noradrenaline on internal models has been relatively sparse, partly due to the requirement of measuring the internal model. In one significant rodent study, increasing noradrenaline results in broader – less selective – tuning curves of auditory neurons (akin to a network reset) (*Martins and Froemke, 2015*). Such an impact on representations is what is required for noradrenaline to implement its proposed role in belief updating (*Nassar et al., 2012*) and increased weighting of current evidence versus prior belief on choice (*Krishnamurthy et al., 2017*). By simultaneously measuring pupil size and an internal model in mOFC, we observed changes in pupil diameter predict increases in uncertainty in an internal model, demonstrating a potential analog of this rodent work in humans performing complex decision-making tasks, and suggesting a broad role of pupil-linked neuromodulation in reducing selectivity of cortical representations.

We then asked which brain regions contain signals associated with changes in the certainty of beliefs and may be involved in regulating the pupil-linked neuromodulatory system. We found ACC tracked changes in the certainty of beliefs, consistent with observations that it monitors the outcomes of one's actions (*Walton et al., 2004*), predicts the updating of beliefs (*Behrens et al., 2007*), and is specifically engaged when internal models need to be updated (*O'Reilly et al., 2013*). ACC was particularly involved when uncertainty should increase, consistent with its role in exploratory behavior (*Hayden et al., 2011*) and information seeking (*Stoll et al., 2016*). While the present study provides evidence for this role of ACC, it was not designed to dissociate it from other roles ascribed to ACC, such as cognitive control (*Shenhav et al., 2013*) and choice difficulty (*Shenhav et al., 2014*). Rather, we double dissociate ACC from other regions representing an internal model (mOFC), and provide evidence ACC may broadcast information to such regions via neuromodulatory systems.

Anatomical work suggests ACC is one of the few cortical regions with descending control on noradrenaline release (*Jodo et al., 1998*). Further, recent studies have shown that activity of ACC neurons predicts pupil dilation and may temporally lead pupil-predicting LC activity (*Ebitz and Platt, 2015*; *Joshi et al., 2016*). Therefore it is anatomically plausible that ACC could influence belief uncertainty via the noradrenaline system, possibly amongst other routes. However, whilst both ACC and noradrenaline have been implicated in belief updating (*Karlsson et al., 2012*; *O'Reilly et al., 2013*) and setting the learning rate (*Behrens et al., 2007*; *Nassar et al., 2012*), evidence for their suspected (*Aston-Jones and Cohen, 2005*; *Karlsson et al., 2012*; *Nassar et al., 2012*) coordination during tasks requiring belief updating has been lacking. Here we found ACC activity explained changes in baseline pupil dilation, over and above the effect of changes in uncertainty and mean brain activity. Whilst our pupil results cannot be linked directly to noradrenaline, they are compatible with the hypothesis that the noradrenaline system could offer a mechanism via which the computation of uncertainty of beliefs in ACC may be broadcast to internal models.

An alternative explanation for the relationship between ACC and pupil dilation is that they are associated, not with a change in beliefs, but with a change in the drive to produce exploratory behavioural output. The circumstances in which model entropy should increase are the same circumstances that should drive participants towards exploratory (random) behavior. Indeed, increasing noradrenaline release to the ACC has been associated with the output of stochastic behavior (*Tervo et al., 2014*). However, an account in which ACC activity and noradrenaline release only track a drive to produce stochastic behavioural output, or to explore, cannot account for the association between pupil dilation and the increase in model uncertainty recorded in mOFC, in the absence of any behavioural change. Further, the animal work described above (*Martins and Froemke, 2015*) provides causal evidence that noradrenaline reduces selectivity of cortical representations, supporting the proposed role for noradrenaline in reducing the strength of beliefs, in addition to its role of increasing stochastic behaviour (*Tervo et al., 2014*). This hypothesis can be summarized in terms of economic models of choice as arguing that noradrenaline input to ACC alters the temperature parameter of a softmax choice model whereas noradrenaline input to internal models (mOFC) alters the option values, resulting in behavioural stochasticity (*Tervo et al., 2014*) and entropic beliefs (present results), respectively.

We started by asking whether there was a mechanism for increasing uncertainty in neural models. The results presented, in which increases in belief uncertainty engage ACC, which explains changes in pupil size, that in turn explain changes to the selectivity of internal models following evidence the underlying state of the world has changed (reward omission), provide a candidate mechanism. These results suggest different neural mechanisms may be engaged when forming as compared to revising beliefs, and that neuromodulatory systems may play a crucial role in implementing flexibility of internal representations during decision-making in humans.

## Materials and methods

### Participants

Twenty-two healthy human volunteers participated in the experiment. Participant identities were anonymized for analyses. All participants were right-handed, aged between 20 and 31 (mean age: 26), and eight were female. All participants had normal or corrected to normal (with contact lenses) vision. Three participants had to be excluded because of technical difficulties with fMRI, and three had to be excluded from pupil analyses because of technical difficulties with eyetracking in the fMRI scanner. Power analyses are as follows. For univariate effects of model updating: Based on a previously published result demonstrating an effect of model updating in the ACC (*O'Reilly et al., 2013*), we performed a simple a-priori power calculation. This effect from *O'Reilly et al. (2013)* had a Cohen's d of 0.89; for this effect size, 80% power and an alpha level of 0.05 (one tailed) the minimum sample size was calculated as 17 subjects. For multivariate effects, no a priori power calculations were performed as effect sizes for multivariate analysis are notoriously hard to estimate in advance, depend on the type of information to be decoded and vary across cortical regions – given that we used a whole brain searchlight approach this would make it very difficult to pre-specify an effect size (*Bhandari et al., 2018*). We present a posteriori power calculations because of a reviewer request. The effect size (Cohen's d) for the key result that representation strength is predicted by model entropy was d = 1.58. For a power of 80% and alpha = 0.05 one tailed, the minimum sample size was calculated at six individuals. Whilst we would not be confident in a result based on such a small sample, we note that the regression analysis did give a negative regression coefficient (high entropy predicts low representation strength) in 18/19 individuals in our group. For comparison, Schuck et al (*Schuck et al., 2016*) decoded task states from a similar medial OFC region to that found in the present study, in an a priori ROI. Their effect size (accuracy of decoding task state vs. chance) was d = 1.77, which would suggest a minimum sample size of 5. For pupillometry effects, the most relevant study in the literature is *Nassar et al. (2012)*. In this study the effect size for a regression of pupil average (similar to our baseline pupil measure) on relative uncertainty yielded an effect size of d = 0.94, which would suggest a minimum sample size of 16 for this analysis. We collected data from a number of participants satisfying the estimated size effects, and in line with other papers in the field. The study was approved by a local University of Oxford ethics committee (ref: MSD-IDREC-C1-2013-171), and all participants gave written informed consent.

## Task

Participants were presented with eight options (*Figure 1a*, *Figure 1—figure supplement 1*), four of which were available for selection in each run of the task (participants completed four runs of the task – see below). The four available options were distinguished from unavailable options by their colour (blue or red, counterbalanced across participants). They were instructed that of the four available options, one had a high and the remainders a low probability of giving rise to reward, if selected. Participants selected, on each trial, any and only one of the four available options by pressing its corresponding button. Following this selection, reward was delivered probabilistically according to whether the option selected was the high probability (70 or 90% probability of reward) or one of the low probability (20%) options, and was displayed on the screen at the time of outcome.

The participants' objective was to accumulate as much reward as possible. Hence, the optimal strategy was to always select the high reward-probability option. This high reward option moved randomly to one of the three other options after a variable number of trials (mean 20 trials, SD 5) and independently of the participants' behaviour – adding to the unpredictability (due to the variable length of the exploratory period). When the high reward option moved, it was randomly selected with equal probability to have a 70 or 90% chance of reward delivery. The task was embedded in a game in which participants controlled a mouse seeking cheese (*Figure 1—figure supplement 1*). Following selection of an option, the mouse moved to the option and the outcome (reward or no reward; cheese or apple, counterbalanced across participants) was revealed and remained on the screen for 200 ms. The next trial began soon after (jittered with a uniform distribution between 0 and 4 s; mean ITI of 3.44 s). The short spacing between trials was necessary, as we required large numbers of trials to drive participants between many periods of exploitation and exploration and because we focused our analysis on a subset of trials. The close trial spacing was also designed to encourage a sense of consistent engagement with a particular strategy within exploit blocks.

Participants completed two sessions – one behavioural and pupillometry only, and one additionally with fMRI – each containing four runs of 200 trials (1600 trials total per participant). The available (blue/red) stimuli switched across runs (see below). Before the first session, participants were given detailed instructions of the task. These included underlying statistics: they were explicitly told the probabilities of reward, although not the true underlying frequency at which the high reward probability jumped. Participants were told the probabilities so they would have comparable understanding of the task when doing the fMRI session. Selecting each of the eight options corresponds to a button press for each of the eight fingers. Before starting the first session, participants were trained on the mappings between the response buttons and the options by doing 300 trials of a training task, whereby one option would be highlighted and they would be required to press the corresponding response button. After this, they performed the four runs of the task in the behavioural session. The following day, participants completed the fMRI session, which similarly required participants to complete four runs of the task. The task and all instructions were the same across the two sessions. Participants were additionally instructed to centrally fixate the fixation cross throughout the whole experiment in both the behavioural and fMRI session.

The motive for having eight options but only four of which were available for selection was to test whether there was any difference between options available for selection but not selected and those not available for selection. The configuration of available options differed across the runs such that all options were available in two of the runs (to allow for training and testing sets of the multivariate model). Therefore the unavailable options were intended to act as a baseline. Analyses testing for differences between available but not selected and unavailable options did not yield significant results (data not shown).

## Modelling of behaviour

In order to generate regressors to analyse neural data, we constructed a normative Bayesian learning model that reflected information communicated to participants – specifically, that one option had a high probability of reward and the remainders a low probability of reward.

We implemented a normative Bayesian learning model to simulate how belief uncertainty should change depending on trial history. The model had information that reflected that which was communicated to participants. We informed participants that there was always and only one location with an either 70 or 90% probability of receiving reward and the remainders only 20%. The model

estimated the probability that each of four hypotheses $H_t \in \{1,2,3,4\}$ is correct – one hypothesis for each of the four locations that could be chosen on any given trial $L_t \in \{1,2,3,4\}$ being the high reward-probability location. In each of these hypotheses, the three low probability locations were assigned p(reward)= 0.2, which was fixed. The other, high reward-probability location was assigned probability P, where P could take any value in the range 0.3 to 1. We allowed P to be more flexible to allow for possible differences in participants' estimates of probabilities, but constrained it to be greater than the default payoff of 0.2 by imposing a lower bound of p=0.3.

This information can be thought of as represented in a payout matrix – a matrix whose elements correspond to a probability of reward being delivered. The payout, or outcome, on trial t, $O_t$, was determined by a binomial distribution parameterised by q:

$$O_t \sim B(q)$$

where q was obtained from the payout matrix M. M depends on which location is the true high reward-probability location H, what the payout probability P at the high reward-probability location is, and which location L was selected on that trial.

The posterior probability that each location was the correct one, H, and the payout probability P takes some value p, was found by considering the payout probability q associated with the cell $(h, L_t, p)$ in the payout matrix:

$$p(H_t = h \cap P_t = p | O_{1:t}, L_{1:t}) = p(q = \mathrm{M}_{h,l,p} | O_{1:t}, L_{1:t})$$

The posterior probability over q, $p(q = \mathrm{M}_{h,l,p} | O_{1:t}, L_{1:t})$, was given using Bayes' rule:

$$p(q = \mathrm{M}_{h,l,p} | O_{1:t}, L_{1:t}) \propto p(O_t | q = \mathrm{M}_{h,L_t,p}) \cdot p(q = \mathrm{M}_{h,L_t,p} | O_{1:t-1}, L_{1:t-1}s)$$

The likelihood $p(O_t | q = \mathrm{M}_{h,l,p}, L_t)$ was simply:

$$p(O_t | q = \mathrm{M}_{h,L_t,p}) = \begin{cases} \mathrm{M}_{h,L_t,p} & \text{for} \quad O_t = 1 \\ 1 - \mathrm{M}_{h,L_t,p} & \text{for} \quad O_t = 0 \end{cases}$$

meaning that the likelihood distribution was a 2D matrix over

$$h = \{1,2,3,4\} \ \times \ p = \{0.3{:}0.01{:}1.0\}$$

The prior $p(q = \mathrm{M}_{h,L_t,p} | O_{1:t-1}, L_{1:t-1}s)$ was none other than the probability of the values H=h and P=p, given all previous observations and the switch probability s:

$$p(q = \mathrm{M}_{h,l,p} | O_{1:t-1}, L_{1:t-1}, s) = p(H_t = h \cap P_t = p | O_{1:t-1}, L_{1:t-1}, s)$$

This was obtained from the previous trial's posterior after accounting for the possibility of a switch in H with a leak. The probability of a switch s was fixed and modelled as 1/20. This reflects the true underlying mean frequency of a switch, but was unknown to participants (we could model this flexibly – however, once in the fMRI session, participants likely have a rough gauge of how frequently there is a switch due to having already completed 800 trials of the task in the behavioural session). The prior was therefore given by:

$$p(H_t = h \cap P_t = p | O_{1:t-1}, L_{1:t-1}, s) = [p(H_t = h \cap P_t = p | O_{1:t-1}, L_{1:t-1}, s) \cdot (1-s)] + p(H_t = h \cap P_t = p | U(\mathrm{M})) \cdot s$$

where $U(\mathrm{M}) \cdot s$ is a uniform leak over the parameter space, weighted by s, allowing the model to 'forget' past outcomes (otherwise the estimates in the posterior will just converge to the mean of all estimates).

To obtain the model entropy over Hypotheses (i.e. the spread of belief over which of the four possible locations was the high reward location), we marginalized over p:

$$p(H = h | O_{1:t}, L_{1:t}, s) = \sum_p p(H = h, P = p | O_{1:t}, L_{1:t}, s)$$

This gave the marginalised posterior shown in *Figure 1e*. We then applied the formula for Entropy:

$$ENT = -\sum_{h=1}^{4} p(H = h | O_{1:t-1}, L_{1:t}, s) \log(p(H = h | O_{1:t-1}, L_{1:t}, s))$$

This was the equation that gave the entropy regressor used in analyses and shown in *Figure 1d*.

## fMRI acquisition and preprocessing

In the fMRI session, four blocks, or runs, of functional data were acquired, with mean ~12 min each. Data were acquired on a 3T Siemens TIM Prisma, using a 64-channel head and neck coil. Functional scans were collected using a gradient echo-planar imaging (EPI) sequence, Multi-band 4, with TR = 1.35 s, TE = 32 ms, flip angle = 74 degrees, and voxel resolution of 2×2×2 mm. Structural scans were acquired using a T1-weighted MP-RAGE sequence, 1×1×1 mm voxels. All scans were of axial orientation angled to ACPC, covering the whole brain.

Preprocessing and univariate analysis of fMRI data was performed using tools from fMRI Expert Analysis Tool (FEAT), part of FMRIB's Software Library (FSL). Data were preprocessed using the FEAT default options. Registration of EPI images to high-resolution structural images and to standard (MNI) space was performed using FMRIB's Linear and Non-Linear Registration Tool, respectively (FLIRT and FNIRT). For GLMs run using FEAT, the four runs of the task for each participant were pooled using fixed effects, and then high-level analyses across participants were performed using mixed effects analyses. For the multivariate and pupil-brain analyses, data was concatenated across runs for each participant and then the GLM was run. These GLMs contained a bias term for each run separately to remove any differences across runs (i.e. the GLM had four bias/main effect regressors, each with ones for one of the runs and zeros for the other three).

## fMRI analyses

For univariate brain GLMs, we locked each trial event to the time of outcome presentation and convolved the BOLD signal with the canonical HRF. The same was performed for generating whole-brain beta maps for the multivariate analyses, which were then used to calculate representation strength (see fMRI multivariate measure of representation strength). We then ran the following GLMs that are presented in the main paper, where the dependent variable was either univariate brain activity, multivariate representation strength, or change in baseline pupil diameter. Analyses were carried out either on all trials or only included trials in the exploit period that were at least five trials from the adjacent explore periods so that any variation is not due to changes in action. These GLMs are specified below:

GLM1 included the following regressors: Exploit, explore. Dependent variable: Univariate brain activity. A [1 -1] contrast tested for differences between exploitation and exploration, with positive activation indicating regions more active in exploitation than exploration, and vice versa for negative activations. All trials included in the analysis. The goal of this analysis was to identify regions that were differentially active in exploitation and exploration phases of the task.

GLM2 included the following regressors: Main effect (bias; column of ones), entropy, whether reward was delivered at outcome, whether participants changed their choice on the subsequent trial, whether participants transitioned from exploitation to exploration on the subsequent trial (last exploit trial), whether participants transitioned from exploration to exploitation on the subsequent trial (first exploit trial). Dependent variable: univariate brain activity. All trials included in the analysis. The goal of this analysis was to identify regions sensitive to certain task factors, as well as to test for regions with activity at the transitions between exploitation and exploration (over and above that explained by the other task regressors, such as switching options).

GLM3 included the following regressors: Exploit, explore, and all the regressors from GLM2 except the main effect regressor. Dependent variable: Univariate brain activity. A [1 -1] contrast tested for differences between exploitation and exploration. All trials included in the analysis. The goal of this analysis was to investigate whether much of the difference between exploitation and exploration was explained by the task factors in GLM2.

GLM4 included the following regressors: Main effect (bias; column of ones), model entropy. Dependent variable: multivariate representation strength. Analysis carried out as a multivariate searchlight analysis. Only exploit trials (>5 trials from adjacent explore phase) included in the

analysis. The goal of this analysis was to identify regions in which the representation strength varied with model entropy.

GLM5 was the same as GLM4 except instead of the entropy regressor used the probability (belief) assigned by the model that the currently exploited option is the high reward option. The goal of this analysis was to show representation strength in the region identified in GLM4 varied with the belief over the currently selected option (not just entropy, which captures the certainty of the belief).

GLM6 included the following regressors: Main effect, change in baseline pupil diameter between the current trial and the next, change in entropy. Dependent variable: change in multivariate representation strength in mOFC (also performed as a whole brain searchlight analysis, as in *Figure 3c*). Analysis performed separately on only exploit trials (>5 from adjacent explore phase), and only explore trials. When this analysis was performed on explore trials, whether participants changed their choice was added as a regressor to the GLM. These analyses were performed separately on trials on which reward was delivered and omitted. The goal of this analysis was to test whether changes in pupil diameter explained changes in representation strength.

GLM7 included the following regressors: Main effect, change in entropy. Dependent variable: Univariate brain activity. Only exploit trials (>5 from adjacent explore phase) included in the analysis. The goal of this analysis was to identify regions in which activity reflected changes in the model entropy, and to test whether they were dissociated from those in which representation strength varied with model entropy (as in GLM4).

GLM8 included the following regressors: Main effect, brain univariate activity, difference in entropy following outcome between current trial and the previous, and whole-brain mean univariate activity. Dependent variable: change in baseline pupil diameter between the current trial and the next. Only exploit trials (>5 from adjacent explore phase) included in the analysis. (When this analysis was performed on all trials, *Figure 3—figure supplement 3*, whether participants changed their choice on the subsequent trial and whether the current trial was the last trial of exploitation were added as regressors to the GLM.) The goal of this analysis was to identify brain regions in which univariate activity explained changes in baseline pupil diameter.

Multiple comparisons correction was performed with cluster mass-based permutation testing with the FSL function randomise. The clusters were thresholded voxelwise at p=0.001 and cluster masses surviving correction at p<0.05 are presented.

## fMRI multivariate measure of representation strength

To obtain a measure of representation strength at each grey matter voxel, we ran a multivariate searchlight analysis over the whole-brain grey matter. We trained and tested the model on different runs of the task. The configuration of available options differed across the runs, such that each option appeared in two different runs. We therefore split the runs in to two pairs – each pair containing all eight options. We train the model on one pair and test on the other, and then repeat, training and testing on the converse pairs.

The data used as input to the multivariate analyses are whole-brain, standard space beta maps for each trial. We then focus our analyses on the cortex using a grey matter mask. At each grey matter voxel, a set of voxels for that searchlight centre is defined by including all voxels within a radius of seven voxels from the searchlight centre and that lie within the grey matter mask. After performing dimensionality reduction on the voxels and ensuring a balanced training set (see following paragraphs), we train a multinomial logistic regression model using the training set and then use this model to obtain probabilities each of the eight locations was selected on each trial of the test set (see below). This is then repeated switching the training and test sets. The result of this analysis is a probability each of the eight options was selected on each trial of the task at each grey matter voxel.

At each searchlight of the multivariate analysis, we performed dimensionality reduction and ensured the number of exemplars of each option in the training set was balanced. We performed dimensionality reduction to denoise the data (*Mante et al., 2013*) and to facilitate training the multivariate model (so that we didn't have many more dimensions [voxels] than exemplars [trials]). We concatenated the data across runs, and then at each searchlight centre extracted an nVoxels x nTrials matrix. To perform the dimensionality reduction, we carried out principle components analysis on the voxels-voxels covariance matrix. This resulted in nVoxels principle components (PCs) –

each a vector of length nVoxels. We then kept only the top twenty PCs, which captured the 20 strongest spatial modes of variation (shared variance across voxels). The eigenvalues associated with the remainder of the PCs were all near 0, suggesting that the 20 PCs we retained captured most of the spatial variance of activity. Having conducted the PCA, we obtained the contributions of each PC on each trial by linear regression of the voxelwise patterns of activity on each trial against the PC vectors. This resulted in a set of 20 PC contributions (the beta weights from the regression) for each trial – a reduced-dimensionality (20 dimensions) representation of the activity on a given trial. These were taken as the dimensions to be used as predictors in fitting the model. The 20 PC weights on each trial were entered as predictor variables in to a multinomial logistic regression with the selected option as the predicted variable to train the model.

We performed dimensionality reduction on the entire dataset (800 trials). Because this data set included both training and test set, a reviewer raised the possibility that the procedure could introduce bias into the pattern classification. Since the PCA is agnostic to trial labels it is not clear how such bias could arise, but to check that the data reduction step did not give rise to spurious results we conducted a permutation analysis in which 1000 sets of null data (obtained by shuffling voxel identities within each trial) were passed through the entire analysis pipeline in the ROI defined on the peak of the OFC effect. The resulting t scores for the regression of representation strength on model entropy (the key result from this analysis) were symmetrically distributed about zero (*Figure 2—figure supplement 2*). The maximum value of t for these 1000 iterations was 3.09, compared to the peak t-score of 6.8 obtained in mOFC. This suggests that our result is not due to bias introduced by the data reduction procedure.

Because subjects had a free choice as to which option they selected, we had variable numbers of trials on which a given participant selected each option (i.e. an imbalanced training set). To avoid introducing bias in to the model to predict stimuli selected more often, we oversampled the training set such that there were equal numbers of trials of each stimulus type used to train the model. We did this by repeating the trials in the training set for each option until all classes had the same number of trials as the most selected option.

After training the model, we used it to provide probabilities of each option being selected on each trial in the remaining data. For each trial in the test data, we used the PC beta weights on that trial as input to the model to obtain the estimated probability that each option was the selected option. This gave a probability on each trial for each of the eight options. To obtain a summary measure, we computed the ratio of the odds of classifying the trial as the chosen option, to the odds of classifying it as one of the offered but unchosen options (i.e. those options that were available for that run of the task but not selected on that trial). We took the logarithm of this ratio, providing us with a measure of the selectivity to the selected action of each searchlight on each trial, namely the log odds ratio, that is in theory normally distributed, although we used non-parametric statistics (permutation testing) to assess the significance of results at the whole brain level. This trial-by-trial scalar measure was then used in subsequent regression analyses to identify regions in which it is explained by participants' belief entropy or baseline pupil diameter.

## Region of interest definitions

The ACC ROIs for the analyses to test whether ACC activity explained changes in representation strength in mOFC were defined by thesholding the activation maps for both GLM7 and GLM8 at p=0.001 and p=0.01 separately. Hence we tried the analysis with four different ROIs separately. For ROIs we extracted the timecourses by applying the inverse of each participant's registration to the mask to project the mask from standard space in to subject space and extracted the mean time series within this region of interest.

## Pupil acquisition and analysis

We simultaneously measured pupil size while participants performed the task, in both the behavioural session and the fMRI session. We used an EyeLink eyetracker, SR Research to acquire pupil size data, sampled at 1000 Hz.

We preprocessed the pupil size data using standard methods. We identified trials on which subjects blinked and removed the blinks by interpolation of values measured just before and just after the identified blinks. Trials on which the pupil measurement was lost were removed from the

analyses. Finally, we removed the first 25 trials of each run of the task from analyses to avoid any luminance changes that may be caused by starting a new run of the task. After removal of these different trials, the number of core exploitation trials went from 201 ± 7.6 to 162 ± 9.4 (mean ± SEM).

Pupil size in all analyses is normalised for each run by expressing it in terms of percentage difference from the overall mean pupil size on that run. Baseline pupil size is defined as mean (normalised) pupil size in the 20 ms window preceding feedback.

# Acknowledgements

We thank Alon Baram, Philipp Schwartenbeck, James Whittington, Emma Lawrence, Anna Shpektor and Jacob Bakermans for very helpful discussions and for comments on an earlier version of this manuscript. We gratefully acknowledge the Medical Research Council (Career Development Award to JXO'R grant MR/L019639/1, and a grant to THM BRT00020) for funding.

# Additional information

### Competing interests

Timothy E Behrens: Senior editor at eLife. The other authors declare that no competing interests exist.

### Funding

| Funder | Grant reference number | Author |
| --- | --- | --- |
| Medical Research Council | MR/L019639/1 | Jill O'Reilly |
| Wellcome Trust | HMR00560.001 | Timothy E Behrens |
| Medical Research Council | BRT00020 | Timothy H Muller |

The funders had no role in study design, data collection and interpretation, or the decision to submit the work for publication.

### Author contributions

Timothy H Muller, Conceptualization, Software, Formal analysis, Investigation, Methodology, Writing—original draft, Writing—review and editing; Rogier B Mars, Resources, Software, Investigation, Methodology, Writing—review and editing; Timothy E Behrens, Conceptualization, Software, Formal analysis, Supervision, Funding acquisition, Methodology, Project administration, Writing—review and editing; Jill X O'Reilly, Conceptualization, Software, Formal analysis, Supervision, Funding acquisition, Investigation, Methodology, Writing—original draft, Project administration, Writing—review and editing

### Author ORCIDs

Timothy H Muller (iD) http://orcid.org/0000-0002-5817-4949
Timothy E Behrens (iD) http://orcid.org/0000-0003-0048-1177

### Ethics

Human subjects: The study was approved by a local University of Oxford ethics committee (ref: MSD-IDREC-C1-2013-171), and all participants gave written informed consent.

### Decision letter and Author response

Decision letter https://doi.org/10.7554/eLife.39404.029
Author response https://doi.org/10.7554/eLife.39404.030

## Additional files

### Supplementary files

• Source code 1. MATLAB code for the Bayesian model in *Figure 1*, and the behavioural datasets to which the model was fit.

DOI: https://doi.org/10.7554/eLife.39404.017

• Transparent reporting form

DOI: https://doi.org/10.7554/eLife.39404.018

### Data availability

1. We have uploaded the brain maps to NeuroVault (URL: https://neurovault.org/collections/4872/). 2. We have uploaded the MATLAB code for the Bayesian model in Figure 1, and the behavioural datasets to which the model was fit. No additional parameters are needed to run this model, all non-free parameters are hard coded and also described in the Supplementary section "Bayesian learning model". The code and data have been uploaded as a source file (Source code 1). 3. We have uploaded the epoched pupil data to Dryad (https://dx.doi.org/10.5061/dryad.jk17vk0), which together with the behavioural data (point 2 above) can be used to replicate pupil analyses. 4. Individual participant data cannot be publicly shared as the ethical approval for our study does not permit this. Due to their large size (>18GB), it's not practical to upload the multivariate fMRI analyses, but these are available on request from the corresponding author. 6. However, for the multivariate analyses, we are able to, and have, uploaded the medial OFC ROI data as a source data file (Figure 2-source data 1). This can be used to replicate the key multivariate results.

The following datasets were generated:

| Author(s) | Year | Dataset title | Dataset URL | Database and Identifier |
|---|---|---|---|---|
| Timothy H Muller | 2019 | Brain maps from: Control of entropy in neural models of environmental state | https://neurovault.org/collections/4872/ | NeuroVault, 4872 |
| Muller T, Mars R, Behrens T, Jill X O'Reilly | 2018 | Data from: Control of entropy in neural models of environmental state | https://doi.org/10.5061/dryad.jk17vk0 | Dryad Digital Repository, 10.5061/dryad.jk17vk0 |

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

# Appendix 1

DOI: https://doi.org/10.7554/eLife.39404.019

## Modelling of beliefs and behaviour

This appendix follows directly from the Modelling of behaviour section in the Methods. It includes additional modelling of participants' behaviour using alternative models.

### Relative Uncertainty

As an alternative measure of belief uncertainty, we calculated relative uncertainty (Nassar et al., 2012). Relative uncertainty is defined as the uncertainty about (variance of) the outcome given the most likely state of the world (defined as joint maximum likelihood estimate of $\{H_t, q_t\}$ divided by uncertainty about the outcome given the full posterior over $\{H_t, q_t\}$.

Over all participants and runs, Relative Uncertainty and Entropy were highly correlated (mean value of Pearson's r over all participants = 0.80, SEM = 0.0056). This reflects the fact that both measures attempt to quantify the same construct (belief uncertainty). The evolution of entropy, relative uncertainty and change point probability (see below) over a representative run is shown in *Appendix 1—figure 1* to give the reader a sense of the similarity between these measures.

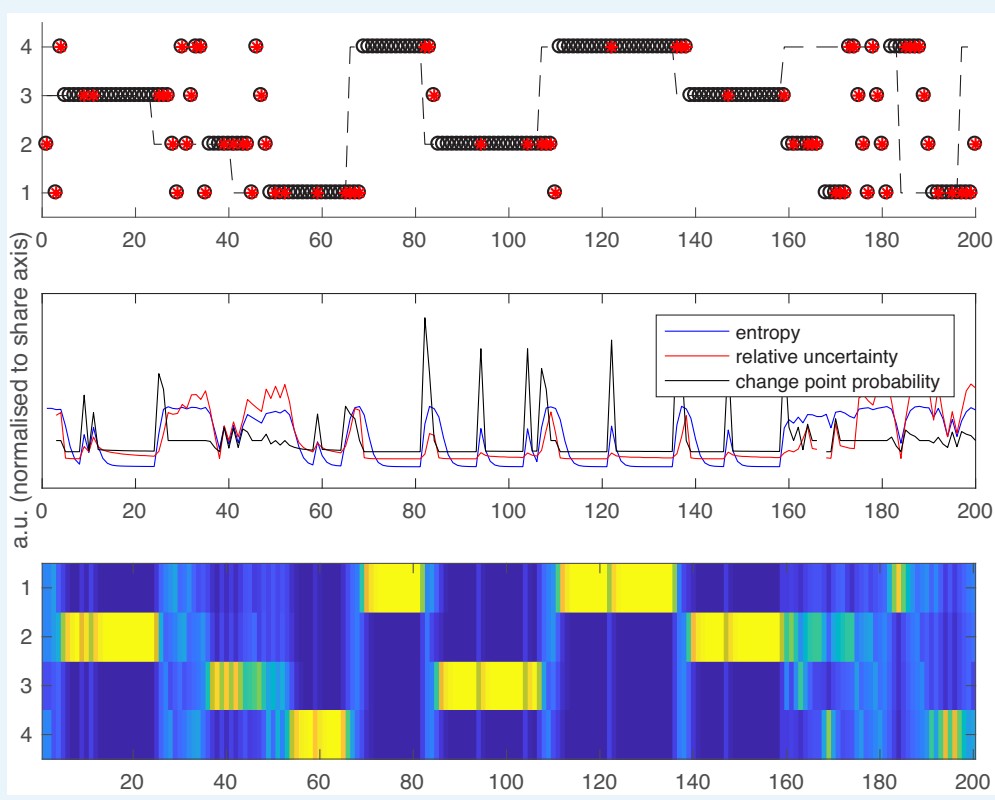

**Appendix 1—figure 1.** Following the same layout as *Figure 1* in the main text. Top panel – an example schedule for one participant/run. Y axis values 1–4 are the possible high reward locations, x axis values are trials. The dashed line shows the ground truth high reward location over time. Open black circles are participants' choices; red dots are reward omissions. Middle panel - Entropy, relative uncertainty and change point probability measures, normalized for comparison. Note that CPP peaks rapidly after reward omission but also falls off rapidly, whilst entropy and relative uncertainty integrate multiple feedback events. Bottom panel – the

probability distribution across candidate high reward locations (bright colors are higher probabilities).

DOI: https://doi.org/10.7554/eLife.39404.020

## Change Point Probability

Change point probability on trial t ($CPP_t$) is defined as the posterior probability that a change occurred between trial t-1 and trial t, after observing data point $x_t$. From Bayes' theorem, this is the probability of observation $x_t$ given that a change point has occurred, multiplied by the prior change point probability (i.e. inverse change point frequency), divided by the probability of observation $x_t$ regardless of whether a change point has occurred.

$$CPP_t = \frac{p(O_t|H_t \neq H_{t-1})p(H_t \neq H_{t-1})}{p(O_t|H_t \neq H_{t-1})p(H_t \neq H_{t-1}) + p(O_t|H_t = H_{t-1})p(H_t = H_{t-1})}$$

In our model this translates as

$$CPP_t = \frac{\left(\frac{3}{4}p(O_t|L_t \neq H_t) + \frac{1}{4}p(O_t|L_t = H_t)\right) \cdot s}{\left(\frac{3}{4}p(O_t|L_t \neq H_t) + \frac{1}{4}p(O_t|L_t = H_t)\right) \cdot s + p(O_t|L_t, H_{t-1}) \cdot (1-s)}$$

As per our Bayesian model, $p(O_t|L_t \neq H_t)$, the probability of the outcome given that there has been a change point and the observer has chosen a low reward location, is given by a binomial with a constant payout parameter (0.2). The probability of the outcome given that there has been a change point and the observer has chosen the high reward location is given by a binomial with the mean payout rate for high reward locations (0.8).

As per Nassar et al (Nassar et al., 2012), for the purposes of calculating CPP we define the probability of the outcome given no change point, $p(O_t|L_t = H_t)$, as the probability of the outcome $O_t$ given that the true state of the world was the joint maximum likelihood values of H and q on trial t-1, i.e. the most likely combination of the identity and payout rate of the high reward option.

Over all participants and all runs, CPP moderately correlated with entropy (mean Pearson's r over all participants 0.37, SEM 0.0055) and weakly correlated with relative uncertainty (mean Pearson's r 0.20, SEM 0.0053). Conceptually, CPP differs from both entropy and relative uncertainty in that CPP measures only the probability of a change point given the current observation, whilst entropy and relative uncertainty integrate information across multiple trials. This can be seen in the example plot of entropy, relative uncertainty and CPP, in which CPP rises and falls on a single trial basis whilst entropy and relative uncertainty vary more smoothly over trials (*Appendix 1—figure 1*).

## Predicting participants' choices

Although the purpose of our modelling was to provide trial-by-trial regressors to relate to neural responses, rather than to optimally capture behaviour, we checked our model was a reasonable fit to behaviour. To determine whether our model was a reasonable fit to the behaviour of our participants, we defined an action policy by fitting a softmax function for each participant and run, in which the model probabilities assigned to each option (of being the high reward option) were used to predict option choice, with softmax temperature as a free parameter. The model's predicted choice (defined as the option with the highest probability under the softmax) matched the participants' actual choices on 94.2% of exploit trials (averaged over all participants; SEM 0.70% across participants) and 31.5% (SEM 1.3%) of explore trials.

Arguably it is more appropriate to fit two separate softmax functions with independent temperature parameters for explore and exploit phases of the task, since behaviour may be qualitatively different in the two cases (for example, behaviour could be more random in the explore phase, which would be reflected in a higher softmax temperature). When we fit softmax functions separately to explore and exploit trials, with temperature as an independent free parameter for each trial category, 96.0% of exploit trials (SEM 0.57%) and 50.9% of explore trials (SEM 1.8%) were predicted correctly. Notably the proportion of explore trials

predicted correctly exceeded the expected proportion if people chose at random during explore phases, which would be 25% (one sample t-test comparing the proportion correctly classified to 25% across participants: $t_{18}$ = 13.2, p=5.2×$10^{-11}$).

We compared the overall model log likelihood for the Bayesian model with a single softmax policy, and two softmax policies (fitted separately to explore and exploit periods) to two further models, which should represent a ceiling in terms of how well the model matches the actual data, in that they are fit to participants' responses rather than to latent variables reflecting participants' modelled beliefs. The first comparison model was a model in which the probability of choosing the selected option was always one during exploit periods and always 0.25 during explore periods. We call this the 'all or none' model because its behaviour is totally deterministic during exploit periods and totally random during explore periods. The second comparison model was a Hidden Markov Model (HMM) described below in which behaviour was classified as 'explore' or 'exploit' based on block length, and then the probability of choosing the selected option was always one during exploit periods and always 0.25 during explore periods as for the all-or-none model. In each case the model log likelihood was defined as the log likelihood of the model making the same choice as the human observer, summed over all trials.

Over all participants, the HMM had the best model log likelihood (mean over all participants = −165), closely followed by the Bayesian model with two separate softmax policies (−179). Both models did much better than the all-or-none model (−308) and the Bayesian model with a single softmax policy (−473). Model log likelihoods for all models are shown in *Appendix 1—figure 2*.

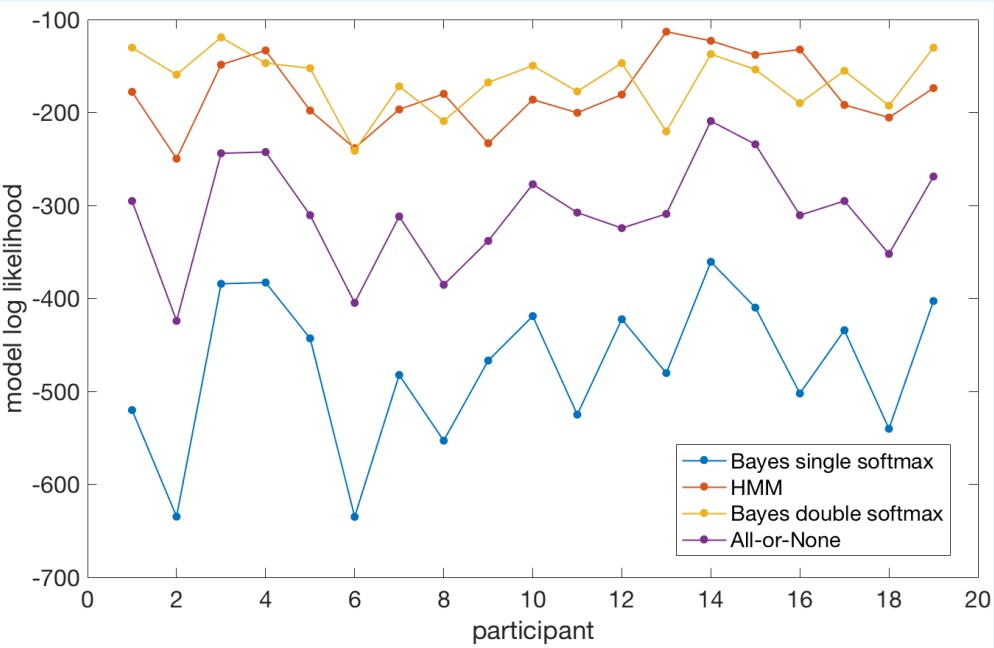

**Appendix 1—figure 2.** Model log likelihoods for each participant and each model, based on predicting participants' choices across all trials. Taken together, the model comparison suggests that the Bayesian model, when allowed to adopt different softmax policies in explore and exploit phases, performs very comparably to a model fitted directly to participants' behaviour (the HMM). Importantly, because the Bayesian model simulates latent states of belief uncertainty (model entropy, relative uncertainty) even when behaviour is held constant during exploitation periods, it provides useful information over and above that given by the HMM (which does not model belief uncertainty at all). A second notable point is that the Bayesian model (with double softmax policy) does predict participants' behaviour during explore periods. In contrast, the all-or-none model (which performed much worse than the Bayesian model with double softmax policy) sets an upper bound on the model log likelihood that

could be achieved if behaviour in the explore period was random (as the probability of the chosen option during exploit periods is one and therefore model log likelihood is maximized in these periods).

DOI: https://doi.org/10.7554/eLife.39404.021

## Does belief uncertainty predict Explore-Exploit transitions?

An important feature of our model based approach was that we were able to model belief uncertainty as entropy or relative uncertainty in the model's hypothesis space. To confirm the validity of this approach, we asked whether high entropy, relative uncertainty or change point probability made participants more likely to transition from exploiting an option to exploration.

Limiting our analysis to trials classified as 'exploit', we used logistic regression to ask whether the probability of switching option on trial t + 1, given that a trial t was unrewarded, depended on the relative uncertainty, entropy or change point probability on trial t (three separate regression analyses). We limited the analysis to cases where trial t was unrewarded, because participants switched chosen option only following unrewarded trials (mean proportion of switch trial following an unrewarded trial = 97.4% across all participants and runs). 21% of all trials over all participants were both classified as exploit and unrewarded and hence used in this analysis.

Both higher entropy and higher relative uncertainty predicted a higher probability of switching (entropy - $t_{18} = 7.60$, $p=2.57 \times 10^{-7}$; relative uncertainty - $t_{18} = 13.50$, $p=3.7 \times 10^{-11}$).

Interestingly, high change point probability on a given trial t predicted a lower probability of switching (change point probability - $t_{18} = 6.45$, $p=2.28 \times 10^{-6}$). The reason for this can be seen in *Appendix 1-figure 3* in which change point probability for the trials in the run up to an exploit-explore transition is plotted. It is evident that CPP actually peaks 2–3 trials before the transition (*Appendix 1-figure 3*). This lag is reflective of the fact that CPP is driven primarily by reward omission on the current trial t, whilst participants' behaviour is guided by integrating outcomes over multiple trials, and thus participants were unlikely to switch on an unrewarded trial, unless other recent trials were also unrewarded. The proportion of exploit-explore transitions for which trial t-1, t-2 and t-3 were unrewarded was (mean ±SEM across participants): 13.1% (±2.36%), 57.5% (±6.4%) and 73.9% (±2,6%) respectively.

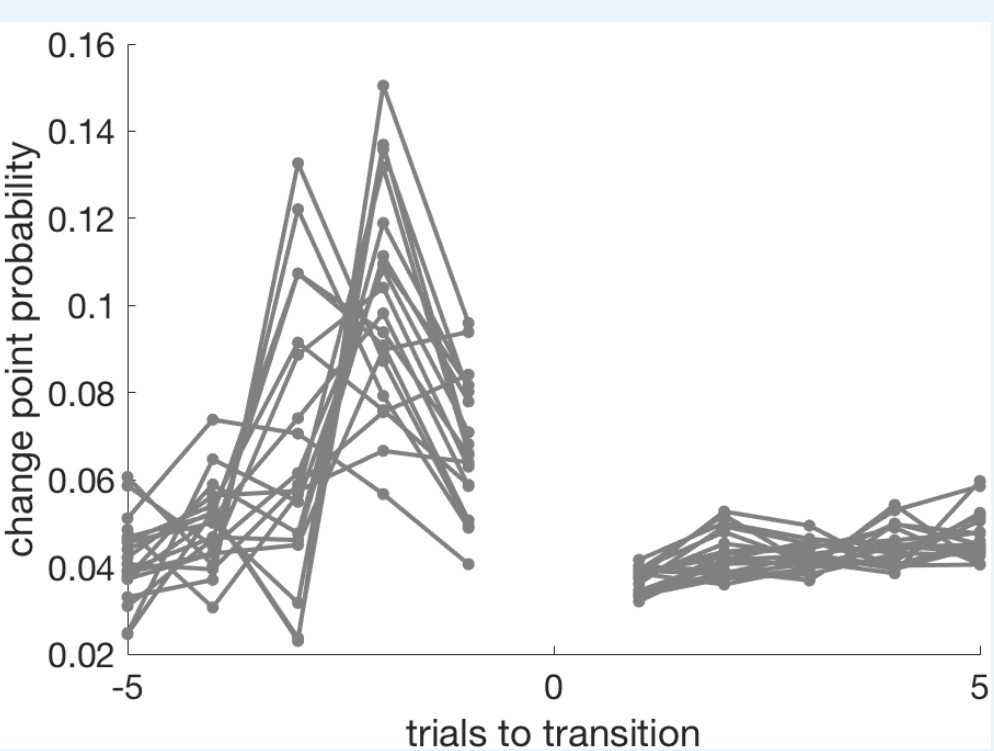

**Appendix 1—figure 3.** Change point probability in the trials leading up to, and following, the transition from exploit- to explore phase. The last exploit trial is coded as −1, the first explore trial as +1 (there is no trial zero in this plot). CPP peaks on the penultimate trial of the exploit block, or the trial before that.

DOI: https://doi.org/10.7554/eLife.39404.022

Both entropy and relative uncertainty are measures of the belief uncertainty that arises from reward omission integrated over multiple trials. To obtain a model-agnostic view of factors predicting exploit-explore transitions, we performed a logistic regression (limited to unrewarded trials in exploit periods) using reward omission on the previous four trials (t-1 to t-4) as explanatory variables. Reward omission on trials t-1 and t-2 significantly predicted option switching (i.e. exploit-explore transitions); on trial t-3 the effect was just n.s. and on trial t-4 the effect was clearly n.s. (p values for trials t-1 to t-4 respectively: $3.81 \times 10^{-6}$, $7.63 \times 10^{-5}$, 0.064, 0.65 – non-parametric sign test used as the regression coefficients were grossly non-normal across the group). These results further confirm that participants integrated reward omission over 3–4 trials (including the final exploit trial, t) to drive exploit-explore transitions.

## Classification of behaviour into Explore and Exploit phases

Throughout the main manuscript we used a heuristic method to define exploitation periods. We defined the start of a period of exploitation as the first trial on which participants selected the true high reward probability option and received reward, and the end of the exploit period as the last trial before switching to a different option (thus within exploit periods, by definition, all choices were for the same option). We furthermore defined 'core exploit' trials as trials within an exploit block, more than five trials from the preceding and following changes of action. The primary aim of this heuristic was to identify, in a conservative way, trials that we could be confident represented stable exploitation behaviour. This was important because we wanted to carry out a number of analyses only on 'core exploit' trials – trials where overt behaviour was stable but belief certainty varied.

It is possible to take a more principled, model-based approach to the definition of explore and exploit phases of the task and we present such an analysis here to justify the use of our heuristic.

Ebitz and colleagues (*Ebitz et al., 2018*) gave careful consideration to how trials may be classified as explore or exploit in a task similar but not identical to the present task (three arm bandit with independently drifting bandit values; the present task is a four arm bandit but the bandit values are tied to each other). They concluded that a Hidden Markov Model (HMM) may be used to classify trials into explore- and exploit- states, based on block length. They fitted a double exponential distribution to block lengths, with the logic that block lengths are shorter in explore periods and longer in exploit periods. They then fitted a hidden Markov model to participants' choices, in which the hidden states were exploit (one state per option) or explore.

We used the same approach to classify trials as explore or exploit in the present task. The hidden states were {exploit option 1, exploit option 2, exploit option 3, exploit option 4, explore}. In each exploit state the only possible observation was the selection of the exploited option. In the explore state all options were equally likely to be selected. Transitions were possible (end equally likely) from the explore state to any exploit state. Transitions were also possible from each exploit state to the explore state, but not directly to another exploit state. The probabilities of transitioning between states were determined by the fitted time constants of the block length distributions for explore- and exploit- states; where $\beta_1$ was the time-constant of the fitted exponential distribution of block lengths in the explore condition, $p_1 = \frac{1}{\beta_1}$ was the probability of transitioning from exploration to an (any) explore state; as all exploit states were equally likely, the probability of transition to any given exploit state was $\frac{1}{4\beta_1}$. Similarly where $\beta_2$ was the time-constant of the fitted exponential distribution of block lengths in the exploit condition, $p_2 = \frac{1}{\beta_2}$ was the probability of transitioning from an exploitation state to the explore state. Transitions directly between exploit states had probability zero.

The classification of trials as 'explore' and 'core exploit' by our heuristic and the Ebitz approach can be compared in *Appendix 1—figure 4*. The critical trial group used for the analysis of stable exploit periods, called 'core exploit', was defined as exploit trials more than five trials from the nearest explore trial. In this trial group the two models were concordant in that almost all trials classified as 'core exploit' by our heuristic were also classified as 'core exploit' by the HMM approach (mean ±SEM across participants: 97.3% ± 0.37%).

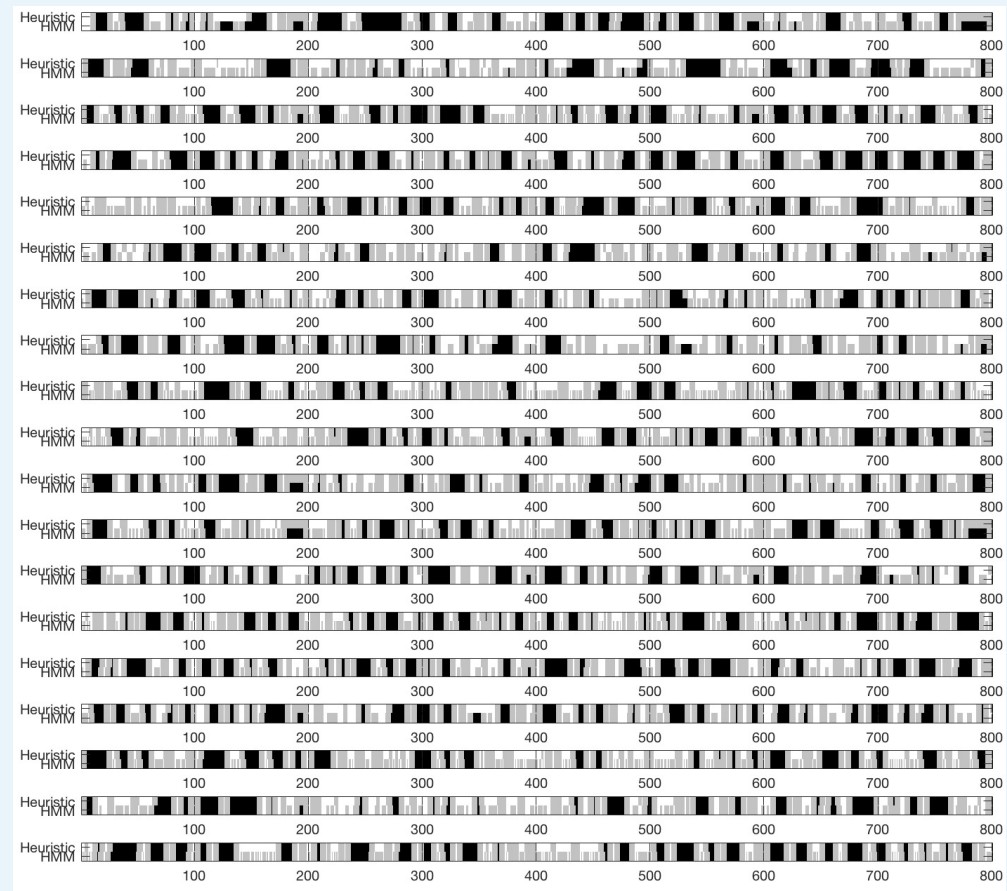

**Appendix 1—figure 4.** Explore and Exploit phases extracted by the Ebitz model. Trials classified as 'exploit' (light grey) and 'core exploit' (dark grey) using our heuristic method (top) and the HMM method (bottom) for each of the 19 participants included in the main analysis. The HMM method tends to classify more trials as 'exploit' but because these are often short blocks, the trials classified as 'core exploit', used in all the multivariate analyses in the paper, are very similar for both classification methods.

DOI: https://doi.org/10.7554/eLife.39404.023

Conversely, the HMM approach classified some additional trials as core exploit, which were not so-classified by our heuristic (mean ±SEM across participants: 7.8% ± 1.3%). By definition, these excess exploit trials arise when participants erroneously select a low reward option for a long period of time, even though it had never (within that exploit phase) been the high reward option. For this reason, these additional exploit trials could be interpreted as periods of off-task behaviour. During the additional exploit trials, model entropy is likely high (as by definition participants are selecting a low-reward option and thus experiencing many reward omissions), but representational strength in any internal model would likely be low (due to disengagement from the task). We thus felt that the inclusion of these additional trials could artificially inflate the relation between model entropy and representation strength and chose to use our more conservative although heuristic approach throughout.

