## [Decision Letter]

[Editors’ note: the authors were asked to provide a plan for revisions before the editors issued a final decision. What follows is the editors’ letter requesting such plan.]

Thank you for sending your article entitled "Control of entropy in neural models of environmental state" for peer review at *eLife*. Your article is being evaluated by Michael Frank as the Senior Editor, Tobias Donner as the Reviewing Editor and three reviewers. The following individuals involved in review of your submission have agreed to reveal their identity: Samuel J Gershman (Reviewer #2).

Given the list of essential revisions, including new experiments, the editors and reviewers invite you to respond within the next two weeks with an action plan and timetable for the completion of the additional work. We plan to share your responses with the reviewers and then issue a binding recommendation.

Summary:

The authors investigate how learners maintain flexibility, a necessary feature of learning in dynamic environments. They propose that noradrenaline increases posterior uncertainty in the currently held representation, thereby maintaining flexibility and counteracting the decrease in uncertainty that is entailed by accumulating more observations. They provide evidence in favor of this hypothesis by means of pupillometry and fMRI. They further propose that the current representation, whose uncertainty is increased by noradrenaline, is encoded in the medial orbitofrontal cortex and that the anterior cingulate cortex detects when the current model is no longer valid and signals it to the locus-coeruleus – noradrenaline system.

Overall, the reviewers found this work interesting. The reported results provide a potentially exciting link between two rapidly evolving domains in computational and cognitive neuroscience. The authors employ some clever analyses; the dissociation between state uncertainty and action variability, in particular, is a nice feature of the analysis.

But all reviewers also raised substantial concerns, of either conceptual or technical nature, which would need to be addressed before this paper can be reconsidered for publication at *eLife*. The conceptual concerns hover around the mechanistic interpretation of the results: reviewers did not feel like the modeling, and the link between model and physiology, were as clear as they could have been.

We realize that addressing all the essential points below might be challenging within the relatively short period of time intended for the revision process at *eLife*, and hence we would like you to prepare a response and action plan.

Essential revisions:

1) Please provide a detailed model-based analysis of participants' behavior, ideally comparing the current model to other plausible ones.

Currently, none of your analyses shows how well the model accounts for participants' behavior. The model doesn't even define a policy, so it doesn't address the explore-exploit trade-off directly. Modeling behavior is important both in order to state clearly a mechanistic hypothesis about choice behavior, and to show empirically that such a model is supported by the data.

Modeling behavior would be critical, for example, to determine whether the representational change in OFC reflects some multi-faceted entropy measure as might be predicted by a "state" account of OFC representation, or over-representation of an error signal during periods of high arousal, which would also be consistent with ideas about value signals in OFC.

By the same token, your primary behavioral figure shows that participants switched more under conditions of high entropy, but this is quite highly correlated with the actual trial outcome. Does the model-based entropy measure affect participant behavior beyond simple win-stay lose-shift or reinforcement learning effects? If so, what are the additional aspects of task environment (e.g. feedback validity, prior feedback history) that jointly affect entropy measures and switching behavior?

2) Please unpack your mechanistic reasoning with regards to entropy. You put forward the idea (i) that posterior uncertainty always decreases with more observations, and therefore (ii) that some specific mechanism must "drive up entropy" in dynamic environments, and they look for such a mechanism in this paper. The reviewers understand this can be an interpretation of a leaky accumulation process, in which the leak "drives up entropy" (ii) and the accumulation "decreases posterior uncertainty" (i). But it is not entirely clear from the perspective of Bayesian inference. Specifically, reviewers raised two issues in this context.

2a) It is not generally true that posterior entropy decreases with more observations.

You state (Introduction): "if beliefs about the environment are expressed as a posterior distribution over possible states, the entropy, or uncertainty, of this posterior decreases as the number of concordant data points increases." Whether or not this is a mathematically correct statement depends on what is meant by "concordant" (which does not seem to be a technical concept). There are plenty of cases where posterior entropy will increase as more data arrive. For example, if you initially strongly believe that a coin is biased to produce heads and then you observe tails, then your posterior will be higher entropy than your prior. (This also comes up later on the same page: "If the 'leak' were removed, the model entropy would still decrease with each new observation, as this is a natural consequence of gaining information").

2b) It is not clear why a dedicated mechanism would be required to "drive up" or "inject" entropy.

Reviewers do not understand why a Bayesian model of inference must "counter" inflexibility by "injecting" entropy. There is nothing, mathematically speaking, that requires a model of (presumably human) inference to inject entropy. Also, talking about mechanisms like "leak" seem to be confusing algorithmic/implementational level concerns with the computational-level analysis of the internal probabilistic model (see point 3a below). The "leak" in the Behrens et al. (2007) model, for example, comes directly from assumptions about volatility; it's not a mechanism that is "injected" into the model to increase entropy. The two computations called (i) and (ii) above are a direct consequence of probability calculus (subsection “Bayesian learning model”). Equation 6 in subsection “Bayesian learning model” shows that the assumption of non-stationarity (s > 0) counterbalances, and may even override, the decrease in posterior uncertainty by constantly injecting a uniform prior in the inference. There is no need to further "drive up entropy". Thus, reviewers feel that the statement in the Introduction: "… injecting uncertainty back into the model's state estimate to make rapid changes of belief possible" needs to be qualified or dropped.

3) There are other mechanistic concepts and related terminology, which should be clarified.

3a) In general, the Introduction tends to conflate the computational level of analysis with the level of the biological mechanisms. Reviewers feel it would help to more carefully separate those two.

3b) In choosing your definition of exploitation, you try to rely on "as few assumptions or arbitrary cut-offs as possible" (Results section). Reviewers don't think that's possible. The definition of exploitation you adopted (the first trial on which participants selected the true high reward probability option and received reward) is arguably just as arbitrary as other sensible definitions. An agent in an exploratory mode will place some probability on this option, and hence its selection does not reliably indicate exploitation. Assumptions are inescapable, and it's better to adopt a model-based definition. But until the policy is defined explicitly, this is not possible.

3c) Reviewers do not understand the statement that uncertainty increases on unrewarded trials (subsection “Noradrenaline as a candidate mechanism for increasing flexibility of belief representations”). It's mathematically possible (though relatively unlikely) that uncertainty will increase on unrewarded trials, if subjects have low confidence during an exploitation phase.

3d) The following statement (Introduction) may be incorrect depending on how one interprets the terminology: "the better established one's model of the environment, the less flexible one's beliefs are." This depends on what "flexible" and "established" mean. If flexibility means something like the degree to which the posterior can deviate from the prior when new data arrive (e.g., expected KL-divergence), and if "established" means lower prior entropy, then one can conceive situations where the claim is incorrect.

4) Please elaborate on the decoding analysis.

4a) You look for a region that shows "stronger [decoded] representation strength for the currently selected option when model entropy is low", which should be a hallmark of a probabilistic model. It would be more natural to look for the posterior distribution of option values, rather than a representation of the selection option. It actually looks like it is the representation decoded in mOFC here. Indeed, if the representation decoded is that of the selected option, then it does not really make sense that the classifier's probability is actually lowest for the current option (compared to other options) in the high entropy condition (Figure 2C). By contrast, this result is expected for the option value. As seen on Figure 1C, when posterior entropy is highest in exploitation periods (the ones analyzed here), the value of the currently selected option tends to be the lowest.

4b) From the same basic perspective, the representational analyses could better clarify the source of the effects, rather than lumping everything into entropy. A difference in decoding reliability driven by differences in error versus correct trials is a rather different effect than one that emerges across feedback-validity conditions when accounting statistically for feedback history. The error versus correct distinction is one where the difference in frequency of these two conditions might lead to artificially low estimates of decoding reliability in error trials, which, given larger pupil responses on error trials, could make interpretation of the OFC pupil results fairly tricky. By this token, one could argue that a number of the findings reported in OFC [general uncertainty, choice-specific belief, and 70% vs 90% in the same regions (subsection “Medial OFC represents probabilistic beliefs about the state of the environment” and Figure 2—supplement 1)] could all be driven by a conjunctive code over feedback and option value, rather than by the tracking of reliability weighted state representations.

5) Please substantiate claims about temporal precedence.

The results are interpreted in terms of temporal precedence with the implicit assumption being that pupil diameter prior to outcome presentation relates to subsequent reliability of OFC representations. However, since pupil diameter, BOLD responses, and entropy all likely contain fairly strong autocorrelations, it would be useful if you could better test this assumption by determining whether the relationships hold even after shifting the pupil and OFC signals in time relative to one another (e.g. does OFC representation reliability predict pupil diameter on subsequent trial?).

6) Please provide a more comprehensive characterization of LC responses, and the methods used to detect them.

You should show the actual response time courses, rather than only statistical maps. You should also show the correlation between the LC and pupil responses. There is a dispute in the literature about the feasibility of LC imaging and the pitfalls and physiological artefacts (e.g. Astafiev, 2010; de Gee et al., 2017). Please also provide more information in Materials and methods section – did you correct for physiological noise (using RETROICOR or related approaches)? If not, it might be better to take the LC findings out altogether and focus on the pupil. Finally, some of your claims seem to require that the effects should specific to the LC-NA system. Substantiating this claim would require contrasting LC responses to responses of other brainstem regions.

7) Please elaborate on the representation of the relevant prior literature.

There is a whole literature on this topic, which is only scarcely referred to. The following references (potentially more) should be included:

– Iigaya, 2016 presents a mechanistic view for driving up entropy with noradrenaline, with the idea that a surprise detection module is used to reset (by means of noradrenaline) the representation currently hold and stored in synapses.

– Meyniel and Dehaene, 2017 for waning and waxing of posterior uncertainty and its neural (fMRI) representation

– Pulcu and Browning, 2017 on pupil size variations with volatility

– Krishnamurthy, Nassar and Gold, 2017 on a link between pupil size and the balance between prior and current evidence in uncertain environments

– de Gee et al., 2017 for showing that fMRI responses in the human LC, but also other neuromodulatory brainstem nuclei, correlate with pupil dilation. In that regard, please tone down the statement that pupil dilation is a specific marker of LC activity (how specific this link is, has rarely been tested directly and the few studies that did, in humans, monkey, and rodents, founds links to several brainstem systems).

[Editors' note: further revisions were requested prior to acceptance, as described below.]

Thank you for resubmitting your article "Control of entropy in neural models of environmental state" for further consideration by *eLife*. Your revised article has been favorably evaluated by a Reviewing Editor, a Senior Editor and three peer reviewers.

The manuscript has been improved but there are some remaining issues that need to be addressed before acceptance, as outlined below.

Summary:

All three reviewers and the Reviewing Editor were overall happy with the revisions and think that the paper has become much clearer. All remain convinced that the paper addresses an important and interesting topic. One reviewer still voiced three major concerns and suggested ways of addressing them through textual revisions, plus one possible additional analysis. We feel that addressing these points through textual revisions, and toning down certain parts of the conclusions, would be essential for publication. We leave it to the authors to decide whether or not perform the additional analysis suggested.

Essential revisions:

1) Mathematical constructs that motivate the study and analysis.

The authors now clearly decompose the inference into (1) computing a posterior distribution and (2) turning this posterior distribution into a predictive distribution by applying a transition-function which, if volatility is assumed, blunts the posterior. Using this (clear) terminology, they then say, "we investigate the hypothesis that the brain has a mechanism for increasing entropy in internal models that is analogous to the transition-function update described above". It seems that their analysis cannot be in line with this, since in their experiment, volatility (the rate of change) is constant, therefore the transition-function is constant, and it is impossible to correlate a brain/pupil signal with it or claim that a brain mechanism has been found for it. We suggest further editing the Introduction clarify this aspect.

2) Effect of entropy on representation strength.

It remains that the decoding results are not fully in line with the cartoon illustration in Figure 1B, in particular, that decoding is not significant in the high entropy condition. Instead, it looks like an effect of value, and indeed the effect is significant in GLM 4 testing reward probability.

We are concerned that the control (point 4, second part, in the response letter) for the contribution of value may not have been adequate. They regressed out, voxel-wise, the linear effect of value. The analysis in the (univariate) residuals does not preclude that there remains multivariate activity related to value, which, if correlated with entropy, will boost the effect of entropy. The conserve approach may be more appropriate: value should be regressed out from entropy, then if the correlation between representation strength (derived from the decoder) and residual entropy remains significant and negative, it will be convincing that the effect is not confounded by value.

We realize that this control analysis may not be possible with the current design (strong negative correlation between value and entropy) – if that is the case, then the claim about a specific involvement of entropy should be toned down.

3) Effect of noradrenaline.

The authors opted for removing the LC-fMRI results about which we had technical concerns. So, their claim about noradrenaline now relies entirely on the pupil data, but this overly specific conclusion is not supported by the available data. Specifically, the authors quote Joshi et al., 2016 to relate baseline pupil size (which is central in their analysis) to noradrenaline levels, but the quoted paper in fact shows that LC activity is specifically related to *changes* in pupil size. Reimer et al., 2016 go further in showing that pupil dynamics is correlated to both, noradrenaline *and* acetylcholine-release, and the former is more strongly related to the fast dilations of the pupil whereas the latter is more strongly related to slow variations in pupil size, which would correspond to fluctuations in baseline pupil size in the current paper. The authors should refrain from linking their results specifically to the LC-NA system; they should link their results to "pupil-linked arousal". They can go on to speculate that the LC-NA system might play a key role in their findings, but then they would need to acknowledge that their slow baseline pupil size effects would be more in line with a cholinergic effect. It does not seem to match well with the influential accounts in this field, but we need to take this recent evidence from careful experiments into neuromodulatory correlates of pupil dynamics seriously.

---

## [Author Response]

[Editors' note: the authors’ plan for revisions was approved and the authors made a formal revised submission.]

Essential revisions:1) Please provide a detailed model-based analysis of participants' behavior, ideally comparing the current model to other plausible ones.Currently, none of your analyses shows how well the model accounts for participants' behavior. The model doesn't even define a policy, so it doesn't address the explore-exploit trade-off directly. Modeling behavior is important both in order to state clearly a mechanistic hypothesis about choice behavior, and to show empirically that such a model is supported by the data.Modeling behavior would be critical, for example, to determine whether the representational change in OFC reflects some multi-faceted entropy measure as might be predicted by a "state" account of OFC representation, or over-representation of an error signal during periods of high arousal, which would also be consistent with ideas about value signals in OFC.By the same token, your primary behavioral figure shows that participants switched more under conditions of high entropy, but this is quite highly correlated with the actual trial outcome. Does the model-based entropy measure affect participant behavior beyond simple win-stay lose-shift or reinforcement learning effects? If so, what are the additional aspects of task environment (eg. feedback validity, prior feedback history) that jointly affect entropy measures and switching behavior?

We carried out the following further behavioural analysis:

a) We added a soft max policy to our model to predict trialwise choice and calculate model log likelihood. This is reported in Appendix 1 subsection “Predicting participants’ choices”, and Appendix 1—figure 2.

b) We implemented the change point model of Wilson and colleagues (requires adapting from the original continuous hypothesis space to the discrete hypothesis space of the current experimental paradigm) to obtain alternative measures of uncertainty (Relative Uncertainty and Change Point probability). Nassar et al., 2012. This is reported in Appendix 1, subsections “Relative uncertainty” and “Change point probability”, and Appendix 1—figures 1, 2, and 3.

c) We implemented the Hidden Markov Model of Ebitz et al. 2018 to obtain an alternative definition of explore/exploit trials and an alternative prediction of trialwise choice. This is reported in Appendix 1 subsection “Classification of behaviour into Explore and Exploit phases” and Appendix 1—figure 4.

d) We did a non model-based (GLM) analysis to determine how well patch leaving (exploit-explore switch) is predicted by recent feedback history (reward omission on trials t-1, t-2 etc) and compared to how well patch-leaving is predicted by entropy. This is reported in Appendix 1 subsection “Does belief uncertainty predict Explore-Exploit transitions?”.

2) Please unpack your mechanistic reasoning with regards to entropy. You put forward the idea (i) that posterior uncertainty always decreases with more observations, and therefore (ii) that some specific mechanism must "drive up entropy" in dynamic environments, and they look for such a mechanism in this paper. The reviewers understand this can be an interpretation of a leaky accumulation process, in which the leak "drives up entropy" (ii) and the accumulation "decreases posterior uncertainty" (i). But it is not entirely clear from the perspective of Bayesian inference. Specifically, reviewers raised two issues in this context.2a) It is not generally true that posterior entropy decreases with more observations.You state (Introduction): "if beliefs about the environment are expressed as a posterior distribution over possible states, the entropy, or uncertainty, of this posterior decreases as the number of concordant data points increases." Whether or not this is a mathematically correct statement depends on what is meant by "concordant" (which does not seem to be a technical concept). There are plenty of cases where posterior entropy will increase as more data arrive. For example, if you initially strongly believe that a coin is biased to produce heads and then you observe tails, then your posterior will be higher entropy than your prior. (This also comes up later on the same page: "If the 'leak' were removed, the model entropy would still decrease with each new observation, as this is a natural consequence of gaining information").2b) It is not clear why a dedicated mechanism would be required to "drive up" or "inject" entropy.Reviewers do not understand why a Bayesian model of inference must "counter" inflexibility by "injecting" entropy. There is nothing, mathematically speaking, that requires a model of (presumably human) inference to inject entropy. Also, talking about mechanisms like "leak" seem to be confusing algorithmic/implementational level concerns with the computational-level analysis of the internal probabilistic model (see point 3a below). The "leak" in the Behrens et al. (2007) model, for example, comes directly from assumptions about volatility; it's not a mechanism that is "injected" into the model to increase entropy. The two computations called (i) and (ii) above are a direct consequence of probability calculus (subsection “Bayesian learning model”). Equation 6 in subsection “Bayesian learning model” shows that the assumption of non-stationarity (s > 0) counterbalances, and may even override, the decrease in posterior uncertainty by constantly injecting a uniform prior in the inference. There is no need to further "drive up entropy". Thus, reviewers feel that the statement in the Introduction: "… injecting uncertainty back into the model's state estimate to make rapid changes of belief possible" needs to be qualified or dropped.

We accept the reviewers’ points and have redrafted the Introduction accordingly. We are not quoting all individual changes here since we did a complete redraft of the Introduction.

3) There are other mechanistic concepts and related terminology, which should be clarified.3a) In general, the Introduction tends to conflate the computational level of analysis with the level of the biological mechanisms. Reviewers feel it would help to more carefully separate those two.

We accept the reviewers’ points and have redrafted the Introduction accordingly. We are not quoting all individual changes here since we did a complete redraft of the Introduction.

3b) In choosing your definition of exploitation, you try to rely on "as few assumptions or arbitrary cut-offs as possible" (Results section). Reviewers don't think that's possible. The definition of exploitation you adopted (the first trial on which participants selected the true high reward probability option and received reward) is arguably just as arbitrary as other sensible definitions. An agent in an exploratory mode will place some probability on this option, and hence its selection does not reliably indicate exploitation. Assumptions are inescapable, and it's better to adopt a model-based definition. But until the policy is defined explicitly, this is not possible.

The Ebitz paper cited above gives careful consideration to how trials may be classified as explore or exploit. In the paper we now present results of defining explore/exploit periods using this HMM approach but decided to keep our heuristic approach for the main analyses as it is actually more conservative.

This is reported in Appendix 1 subsection “Classification of behaviour into Explore and Exploit phases” and Appendix 1—figure 4.

3c) Reviewers do not understand the statement that uncertainty increases on unrewarded trials (subsection “Noradrenaline as a candidate mechanism for increasing flexibility of belief representations”). It's mathematically possible (though relatively unlikely) that uncertainty will increase on unrewarded trials, if subjects have low confidence during an exploitation phase.

We have edited this sentence to make clear that it is possible, but unlikely, that uncertainty would increase on rewarded trials.

3d) The following statement (Introduction) may be incorrect depending on how one interprets the terminology: "the better established one's model of the environment, the less flexible one's beliefs are." This depends on what "flexible" and "established" mean. If flexibility means something like the degree to which the posterior can deviate from the prior when new data arrive (e.g., expected KL-divergence), and if "established" means lower prior entropy, then one can conceive situations where the claim is incorrect.

We accept the reviewers’ points and have redrafted the Introduction accordingly. We are not quoting all individual changes here since we did a complete redraft of the Introduction.

4) Please elaborate on the decoding analysis.4a) You look for a region that shows "stronger [decoded] representation strength for the currently selected option when model entropy is low", which should be a hallmark of a probabilistic model. It would be more natural to look for the posterior distribution of option values, rather than a representation of the selection option.

We said in our response plan that we would attempt to decode the full posterior from mOFC.

However, when we came to do this analysis, we found that within core exploit trials (the trials that went into the fMRI analysis) it was not possible to dissociate the model probability associated with the three unchosen options from each other or from the probability associated with the chosen option. This is because on core exploit trials, by definition none of the unchosen options have been sampled for at least 5 trials. Therefore, most of the evidence informing the probabilities assigned to these options is in fact based on the sampling of the chosen option (i.e., if the chosen option is less likely to be correct based on recent feedback, all other options are upweighted equally). The average correlation matrix between the model probabilities for chosen option and the three other options (ordered from most to least probable under the model) is given in Author response image 1. As can be seen from this matrix, decoding the probability that the chosen option is correct is virtually equivalent, under the model, to decoding the full posterior over all 4 options. This is an inescapable feature of the design of our task.

The correlation matrix shows that the probability assigned to the exploited option is anticorrelated with that assigned to each of the other three, which themselves are all highly correlated.

It actually looks like it is the representation decoded in mOFC here. Indeed, if the representation decoded is that of the selected option, then it does not really make sense that the classifier's probability is actually lowest for the current option (compared to other options) in the high entropy condition (Figure 2C). By contrast, this result is expected for the option value.

To address the question, we repeated the decoding analysis (of option value) after having regressed out the effect of value, and the effect of recent feedback (reward omission on the current trial and previous 4 trials) from the signal at each voxel in the mOFC ROI. We were still able to decode chosen option identity from mOFC and the representation strength was proportional to model entropy.

We would like to point out, and have clarified in the manuscript, that we are not claiming that value/ feedback are not represented in mOFC. Indeed, model entropy is closely related to both option value and feedback history. The point is that we attempted to decode the *identity* of the chosen option (and hence, the current state of the environment) from mOFC, and found that entropy/value/feedback history modulated the strength with which one state representation dominated the activity pattern. Thus, the pattern of activity in mOFC represents a conjunctive code of state and confidence in that state, or state and value of that state, which are closely correlated in the current paradigm (as they would be in most RL paradigms).

Representation strength in mOFC was better predicted by model entropy than by the value of the chosen option; however, these two measures were rather closely correlated and were both related to recent feedback history. Because model entropy and option value both depended, within our learning model, on feedback history, this is to be expected; we cannot distinguish between these conceptually and statistically related constructs.

As seen on Figure 1C, when posterior entropy is highest in exploitation periods (the ones analyzed here), the value of the currently selected option tends to be the lowest.

Note that in Figure 2C, the probability of decoding the current option is not actually significantly below chance, we think it is around chance and appears below chance due to noise.

4b) From the same basic perspective, the representational analyses could better clarify the source of the effects, rather than lumping everything into entropy. A difference in decoding reliability driven by differences in error versus correct trials is a rather different effect than one that emerges across feedback-validity conditions when accounting statistically for feedback history. The error versus correct distinction is one where the difference in frequency of these two conditions might lead to artificially low estimates of decoding reliability in error trials, which, given larger pupil responses on error trials, could make interpretation of the OFC pupil results fairly tricky. By this token, one could argue that a number of the findings reported in OFC [general uncertainty, choice-specific belief, and 70% vs 90% in the same regions (subsection “Medial OFC represents probabilistic beliefs about the state of the environment” and Figure 2—supplement 1)] could all be driven by a conjunctive code over feedback and option value, rather than by the tracking of reliability weighted state representations.

We asked for clarification on this point and it was agreed that the analyses described in response to the previous point should address this point also.

5) Please substantiate claims about temporal precedence.The results are interpreted in terms of temporal precedence with the implicit assumption being that pupil diameter prior to outcome presentation relates to subsequent reliability of OFC representations. However, since pupil diameter, BOLD responses, and entropy all likely contain fairly strong autocorrelations, it would be useful if you could better test this assumption by determining whether the relationships hold even after shifting the pupil and OFC signals in time relative to one another (eg. does OFC representation reliability predict pupil diameter on subsequent trial?).

We have repeated the pupil-representation strength analysis shifting the trials used in the analysis by one trial either way (so pupil on trial t is paired with representation strength on trial t+1 or t-1). The analysis does not yield significant results if we make these shifts, suggesting the association is relatively specific to each trial.

This is reported in Results section of the manuscript, statement beginning “To test the temporal specificity of the result…”

We should point out that we are not trying to make a strong claim about temporal precedence. We are simply saying that pupil dilation changes and representation strength changes tend to co-occur on the same trial as each other, as would be expected if internal models are updated on a trial-by-trial basis.

6) Please provide a more comprehensive characterization of LC responses, and the methods used to detect them.You should show the actual response time courses, rather than only statistical maps. You should also show the correlation between the LC and pupil responses. There is a dispute in the literature about the feasibility of LC imaging and the pitfalls and physiological artefacts (e.g. Astafiev, 2010; de Gee et al., 2017). Please also provide more information in Materials and methods section – did you correct for physiological noise (using RETROICOR or related approaches)? If not, it might be better to take the LC findings out altogether, and focus on the pupil. Finally, some of your claims seem to require that the effects should specific to the LC-NA system. Substantiating this claim would require contrasting LC responses to responses of other brainstem regions.

We have removed this analysis entirely on the reviewer’s recommendation.

7) Please elaborate on the representation of the relevant prior literature.There is a whole literature on this topic, which is only scarcely referred to. The following references (potentially more) should be included:– Iigaya, 2016 presents a mechanistic view for driving up entropy with noradrenaline, with the idea that a surprise detection module is used to reset (by means of noradrenaline) the representation currently hold and stored in synapses.– Meyniel and Dehaene, 2017 for waning and waxing of posterior uncertainty and its neural (fMRI) representation– Pulcu and Browning, 2017 on pupil size variations with volatility– Krishnamurthy, Nassar and Gold, 2017 on a link between pupil size and the balance between prior and current evidence in uncertain environments– de Gee et al., 2017 for showing that fMRI responses in the human LC, but also other neuromodulatory brainstem nuclei, correlate with pupil dilation. In that regard, please tone down the statement that pupil dilation is a specific marker of LC activity (how specific this link is has rarely been tested directly and the few studies that did, in humans, monkey, and rodents, founds links to several brainstem systems).

Thanks for pointing these references out, we have now included discussion of these in the manuscript.

[Editors' note: further revisions were requested prior to acceptance, as described below.]

Essential revisions:1) Mathematical constructs that motivate the study and analysis.The authors now clearly decompose the inference into (1) computing a posterior distribution and (2) turning this posterior distribution into a predictive distribution by applying a transition-function which, if volatility is assumed, blunts the posterior. Using this (clear) terminology, they then say, "we investigate the hypothesis that the brain has a mechanism for increasing entropy in internal models that is analogous to the transition-function update described above". It seems that their analysis cannot be in line with this, since in their experiment, volatility (the rate of change) is constant, therefore the transition-function is constant and it is impossible to correlate a brain/pupil signal with it or claim that a brain mechanism has been found for it. We suggest further editing the Introduction clarify this aspect.

We agree with the reviewer’s point here: indeed, we have not manipulated the magnitude of the transition function. This is in contrast to many previous papers investigating uncertainty, which have effectively searched for neural correlates of a control process (that determines the level of uncertainty) (discussed in the Introduction, fifth paragraph). The approach used by these previous papers is to manipulate the environment so that the optimal level of uncertainty in the transition function varies over time. To introduce this idea, we need to explain the concept of the transition function.

However, in the current study our aim was not to manipulate the optimal level of uncertainty in a model and identify brain regions tracking this, but to measure uncertainty as an intrinsic property of a model of the environment (discussed in the Introduction).

We tried to make the distinction between these concepts clear in the part of the Introduction referring to “2 questions” (fifth to final paragraph). However, we agree that the text suggested we had actually manipulated the transition function, rather than studied how uncertainty within a probabilistic world model is controlled – which amongst other things provides a mechanism by which a transition function could be implemented.

To try to clarify this point we have added a paragraph as follows

“In the current experiment we focused on the second question above [i.e. not the strength of the leak but its consequences as manifest in the internal model itself]. […] These are key components for any candidate mechanism for the implementation of a transition-function that increases uncertainty, such as a leak.”

2) Effect of entropy on representation strength.It remains that the decoding results are not fully in line with the cartoon illustration in Figure 1B, in particular, that decoding is not significant in the high entropy condition.

We think the reviewer is referring to the mOFC results in Figure 2C here. This figure illustrates the effect of entropy on decoding accuracy and decoded option probabilities using a median split of trials into high and low entropy. It is true that the figure suggests that decoding would not be significant in this subset of trials. This simply reflects the key result that decoding was (parametrically) more accurate when entropy was lower. This graded effect is the effect we have tested statistically (analysis described in subsection “Decoding a probabilistic representation of beliefs about the state of the environment”).

The cartoon in Figure 1B is just an illustrative cartoon, which is supposed to show that the effect of entropy on representation strength is graded (i.e. we would expect the high reward option to be represented more strongly than other options in general, but that this pattern would be weaker in high entropy conditions). This is compatible with the linear regression showing a relationship between entropy and representation strength (analysis described in subsection “Decoding a probabilistic representation of beliefs about the state of the environment”). We don’t think it would make the cartoon more informative to change it to look like the median split results when in fact the effect of entropy on representation strength is graded.

Instead, it looks like an effect of value, and indeed the effect is significant in GLM 4 testing reward probability. We are concerned that the control (point 4, second part, in the response letter) for the contribution of value may not have been adequate. They regressed out, voxel-wise, the linear effect of value. The analysis in the (univariate) residuals does not preclude that there remains multivariate activity related to value, which, if correlated with entropy, will boost the effect of entropy. The conserve approach may be more appropriate: value should be regressed out from entropy, then if the correlation between representation strength (derived from the decoder) and residual entropy remains significant and negative, it will be convincing that the effect is not confounded by value.We realize that this control analysis may not be possible with the current design (strong negative correlation between value and entropy) -- if that is the case, then the claim about a specific involvement of entropy should be toned down.

The reviewer is correct that there is a strong relationship between the expected value of choosing the currently selected option, and the ideal observer’s confidence that this is the correct option. Therefore, it doesn’t make sense to regress out option value from entropy – even if there was a residual relationship between entropy and representation strength, it is not clear what the residual of entropy, after value is regressed out, reflects conceptually in this context. In our opinion this is not so much a bug as a feature – because a relationship between action’s expected value, and the agent’s certainty that is the correct action to take, would be a property of many naturalistic models of value based action. However, we agree that this should be spelled out clearly and have added a paragraph to do so, at an early point in the paper (directly after we introduce the idea of model entropy itself).

This is the text we added (subsection “What is the nature of the representation in mOFC?”):

“Throughout the next sections of this paper, we discuss the relationship between model entropy (in our theoretical model) and the level of uncertainty in a neural model of the state of the world. […] In behavioural control more generally, this association would hold for any world model for which the aim was to determine the expected value associated with different candidate actions.”

We have retained the text about the control analysis (in which we regress out the univariate effect of value) because we think it may be of interest to readers in addition to the text above, and made the rest of the paragraph clearer:

“The reason we interpret the multivariate results as evidence for a probabilistic representation of the current state (which option is the current high reward option) is that our ability to decode the identity of the chosen option from mOFC (indexed as representation strength for the identity of that option) was higher when the participant’s uncertainty (model entropy) was lower. […] Rather the decoding must be driven by different relative weightings of specific option representations, consistent with a probabilistic state representation.”

3) Effect of noradrenaline.

*The authors opted for removing the LC-fMRI results about which we had technical concerns. So, their claim about noradrenaline now relies entirely on the pupil data, but this overly specific conclusion is not supported by the available data. Specifically, the authors quote Joshi et al., 2016 to relate baseline pupil size (which is central in their analysis) to noradrenaline levels, but the quoted paper in fact shows that LC activity is specifically related to* changes *in pupil size. Reimer et al., 2016 go further in showing that pupil dynamics is correlated to both, noradrenaline* and *acetylcholine-release, and the former is more strongly related to the fast dilations of the pupil whereas the latter is more strongly related to slow variations in pupil size, which would correspond to fluctuations in baseline pupil size in the current paper. The authors should refrain from linking their results specifically to the LC-NA system; they should link their results to "pupil-linked arousal". They can go on to speculate that the LC-NA system might play a key role in their findings, but then they would need to acknowledge that their slow baseline pupil size effects would be more in line with a cholinergic effect. It does not seem to match well with the influential accounts in this field, but we need to take this recent evidence from careful experiments into neuromodulatory correlates of pupil dynamics seriously.*

We agree that the pupillometry effects alone can’t be linked directly to noradrenaline as opposed to Ach, for example. We have edited the language throughout to be careful about whether we are reporting (a) our results which speak only to pupil-related arousal per se, (b) other groups’ results, some of which are in fact specific to noradrenaline (animal studies, anatomical studies, etc) or (c) other researchers’ theories, which are specifically theories about noradrenaline, even if experimental evidence for a specific role of noradrenaline as described in said theories is still lacking. In other words, we have removed any claim that our results specifically index noradrenaline, but we have not in general removed specific references to noradrenaline where discussing the work of others.

We have also added some text to make clear the current state of the literature, as we understand it, about how (non)specifically pupil data can be linked to noradrenaline:

“We used baseline pupil size (mean pupil area in the 20ms before outcome presentation, expressed as% signal change relative to the participant’s mean pupil area over the task run) to index neuromodulatory state. […] However, we must emphasise that pupillometry is a very indirect measure of neuromodulation and although we mostly interpret our results in the context of noradrenaline due to the extensive literature on pupillometry and noradrenaline, as well as influential theories and experimental work linking noradrenaline to uncertainty, model updating and exploration, we note that our pupil size data cannot be specifically linked to noradrenaline over other pupil-linked arousal factors such as acetylcholine.”